# SARS-CoV-2 suppresses IFNβ production mediated by NSP1, 5, 6, 15, ORF6 and ORF7b but does not suppress the effects of added interferon

Maya Shemesh[1©], Turgut E. Aktepe[2©], Joshua M. Deerain[2], Julie L. McAuley[2], Michelle D. Audsley[3], Cassandra T. David[3], Damian F. J. Purcell[2], Victoria Urin[1], Rune Hartmann[4], Gregory W. Moseley[3], Jason M. Mackenzie[2], Gideon Schreiber[1]*, Daniel Harari[1]*

**1** Department of Biomolecular Sciences, Weizmann Institute of Science, Rehovot, Israel, **2** Department of Microbiology and Immunology, University of Melbourne, at The Peter Doherty Institute for Infection and Immunity, Parkville, Melbourne, Victoria, Australia, **3** Department of Microbiology, Biomedicine Discovery Institute, Monash University, Victoria, Australia, **4** Department of Molecular Biology and Genetics, Aarhus University, Aarhus, Denmark

© These authors contributed equally to this work.
* daniel.harari@weizmann.ac.il (DH); gideon.schreiber@weizmann.ac.il (GS)

## Abstract

Type I Interferons (IFN-Is) are a family of cytokines which play a major role in inhibiting viral infection. Resultantly, many viruses have evolved mechanisms in which to evade the IFN-I response. Here we tested the impact of expression of 27 different SARS-CoV-2 genes in relation to their effect on IFN production and activity using three independent experimental methods. We identified six gene products; NSP6, ORF6, ORF7b, NSP1, NSP5 and NSP15, which strongly (>10-fold) blocked MAVS-induced (but not TRIF-induced) IFNβ production. Expression of the first three of these SARS-CoV-2 genes specifically blocked MAVS-induced IFNβ-promoter activity, whereas all six genes induced a collapse in IFNβ mRNA levels, corresponding with suppressed IFNβ protein secretion. Five of these six genes furthermore suppressed MAVS-induced activation of IFNλs, however with no effect on IFNα or IFNγ production. In sharp contrast, SARS-CoV-2 infected cells remained extremely sensitive to anti-viral activity exerted by added IFN-Is. None of the SARS-CoV-2 genes were able to block IFN-I signaling, as demonstrated by robust activation of Interferon Stimulated Genes (ISGs) by added interferon. This, despite the reduced levels of STAT1 and phospho-STAT1, was likely caused by broad translation inhibition mediated by NSP1. Finally, we found that a truncated ORF7b variant that has arisen from a mutant SARS-CoV-2 strain harboring a 382-nucleotide deletion associating with mild disease (Δ382 strain identified in Singapore & Taiwan in 2020) lost its ability to suppress type I and type III IFN production. In summary, our findings support a multi-gene process in which SARS-CoV-2 blocks IFN-production, with ORF7b as a major player, presumably facilitating evasion of host detection during early infection. However, SARS-CoV-2 fails to suppress IFN-I signaling thus providing an opportunity to exploit IFN-Is as potential therapeutic antiviral drugs.

**Data Availability Statement:** All relevant data are within the manuscript and its Supporting Information files.

**Funding:** We gratefully acknowledge grant support from the following agencies: For the Weizmann Institute, Israel (Schreiber Lab): The Israel Science Foundation (grant No. 3814/19) within the Kill Corona – Curbing Coronavirus Research Program (GS). The Weizmann Institute further acknowledges a grant provided by the Ben B. and Joyce E. Eisenberg Foundation (GS). For The University of Melbourne, Australia: A grant from the Jack Ma Foundation (DFJP) and grants administered by the State Government of Victoria (JMM) to the Mackenzie and to the Purcell Labs. For Monash University, Australia (Moseley Lab): National Health & Medical Research Council, Australia grants: NHMRC Project #1160838 (GWM) and NHMRC Project #1125704 (GWM). The funders had no role in study design, data collection and analysis, decision to publish, or preparation of the manuscript.

**Competing interests:** The authors have declared that no competing interests exist.

## Author summary

Our immune system provides the first-line of defense to combat viral infection through the localized triggering of Type I Interferons (IFN-Is). This is a family of cytokines which send a localized "red alert" signal to both infected and adjacent cells, activating hundreds of antiviral genes thus triggering a major arm of innate immunity. So important are IFN-Is, that it is perhaps not surprising that many viruses have evolved mechanisms to block the triggering of IFN-I production and sometimes to additionally block IFN-I signaling. Herein we have demonstrated that SARS-CoV-2, the virus responsible for COVID19, harbors six genes that block the production of IFNβ, a major subtype of IFN-Is. However, even though SARS-CoV-2 infection had the capacity to strongly decrease pSTAT1, a lynchpin molecule required to disseminate IFN-I signaling, enough residual pSTAT1 remained to allow activation of the anti-viral program. Our findings demonstrate that this coronavirus has devoted a significant proportion of its genome to block IFN-I production, presumably in order to help it establish early stage infection. Nevertheless, this virus remains sensitive to the effects of added interferon, providing an opportunity to treat COVID19 patients with IFN-Is therapeutically.

## Introduction

Type I interferons (IFN-I)s are a family of cytokines first identified in 1957 by their ability to "interfere" with influenza virus replication and dissemination [1]. A cell detects viral infection by pattern recognition receptors, which signal through one of three independent mechanisms–STING (Stimulator Of Interferon Response cGAMP Interactor), MAVS (Mitochondrial Antiviral Signaling Protein) or some members of the TLR (Toll-Like-Receptor) family. These elicit the secretion of IFN-Is into the local microenvironment, triggering innate immunity [2]. An infected cell thus induces localized secretion of IFN-I into the microenvironment promoting signaling in surrounding infected and non-infected cells by binding to cognate cell-surface co-receptors, IFNAR1 and IFNAR2. This triggers the phosphorylation of associated STATs (Signal Transducer and Activators of Transcription), which activate the transcription of hundreds of Interferon stimulated genes (ISGs), eliciting a programmed anti-viral response (reviewed by [3–5]). IFN-I-activated cells trigger innate and adaptive immunity by multiple mechanisms, including upregulation of MHC-I, other cytokines and chemokines—promoting attraction of immune cells to the site of infection, including a pro-inflammatory T-Helper type 1 (Th1) response, enhancement of APC/T-cell cross priming, and augmentation of humoral immunity [6–8]. So important are the anti-viral effects of IFN-Is that viruses often encode genes which function to block IFN-I production or signaling [4]. This is even the case for viruses with extremely small genomes, such as influenza [9], Ebola [10], rabies [11] and mumps [12]. Blockade of IFN-I signaling/production was also reported for multiple genes of the coronaviruses SARS-CoV [13,14] and MERS-CoV [15,16]. For SARS-CoV, both the production of IFNβ and activation of IFN-I signaling was reported to be attenuated. SARS-CoV NSP5 protein was shown to interact with ISG15 and to antagonize the IFN-I-mediated antiviral response [17]. In addition, SARS-CoV infection inhibited activation of IRF3/7, reducing IFNβ production [18]. SARS-CoV ORF6 protein also blocked expression of STAT1-activated genes [13]. In contrast, MERS-CoV ORF4b was shown to antagonize IFNβ production by inhibiting IRF3 and IRF7 [19]. SARS-CoV and MERS-CoV also encode a papain-like protease that impeded immune response [20].

Interestingly, while the antiviral potency of IFN-Is on SARS-CoV is only moderate, SARS-CoV-2 seems to be highly sensitive to exogenously administered IFN-I, indicated by a significant reduction in viral replication following IFN-I treatment [21,22]. Moreover, IFN-I inhalation therapy (but not necessarily systemic injection) was shown to provide promising clinical response to COVID-19 patients [23–26]. This raises the question as to why the innate immune system fails to effectively combat SARS-CoV-2? In recent months, a number of publications have attempted to address this question. A consensus support that SARS-CoV-2 genes can block the production of IFNβ. To identify the molecular mechanisms that block IFNβ production through activation of IRF3/7, cells were transfected individually with CoV-2 viral genes as well as either RIG-I, MDA5 or MAVS to induce IFNβ production [27–29]. Using the same IFNβ-luciferase reporter and RIG-I, Yuen et al identified NSP13, NSP14, NSP5, and ORF6 as blockers of IFNβ production, while Lei et al identified NSP1, NSP3, NSP12, NSP14, ORF3, ORF6 and the M protein, and Xia et al identified NSP1, NSP6, NSP13 and ORF6. In addition to the above-mentioned SARS-CoV-2 genes, ORF3b was implicated by Konno et al as being a potent antagonist towards IFN-I production [30].

In addition to blocking IFNβ production, some reports indicated that SARS-CoV-2 proteins block IFN-I signaling as well. Using an interferon-stimulated response element (ISRE)-luciferase promoter assay, Yuen et al reported that ORF6 blocks reporter activity after stimulation with IFN-I [27], while Lei et al reported multiple genes to block this—NSP1, NSP3, NSP13, NSP14, ORF6, ORF8 and the N protein [28]. Using a similar setup, Xia et al reported that IFN-I signaling to be blocked by NSP1, NSP6, NSP13, ORF3a, ORF6, ORF7b and the M protein [29]. ORF9b was suggested to antagonize Type I and III IFNs by targeting multiple components of RIG-I/MDA-5-MAVS, TLR3-TRIF, and cGAS-STING signaling pathways [31,32]. ORF6 was reported to disrupt nucleo-cytoplasmic transport through interactions with Rae1 and Nup98, thereby inhibiting STAT nuclear import [33,34]. NSP14 was shown to reduce endogenous IFN receptor levels and ORF3a and ORF7a to perturb the late endosomal/trans-Golgi network [35]. The papain-like protease, which is essential for viral polyprotein processing, was shown to catalyze the cleavage of the ubiquitin-like modifier ISG15, which is an IFN-I induced gene and homeostatic regulator of IFN-I signaling [36]. The membrane glycoprotein M was suggested to interact with MAVS, impairing MAVS aggregation and recruitment of downstream TRAF3, TBK1, and IRF3, leading to attenuation of the innate antiviral response [37]. In summary, multiple reports provided alternative and often conflicting identities of coronavirus genes that are responsible for suppression of IFN-I production by SARS-CoV-2. The same confusion relates to the identification (if indeed relevant) of SARS-CoV-2 genes that suppress IFN-I signaling (for review see [38]).

Herein we describe the mapping of SARS-CoV-2 genes that contribute to the suppression of MAVS-induced IFNβ production or the blocking of IFNβ signaling. We employed a multi-levelled approach to test for IFN-I production, inclusive of promoter assays, measurements of mRNA levels and finally measurement of IFNβ protein levels secreted into the media from MAVS-induced HEK-293T cells co-transfected with a vector expressing individual SARS-CoV-2 genes. We then correlated our observations from individual gene transfections with those obtained using SARS-CoV-2 virus infection. Our data show that six viral genes are responsible for strong (>10-fold) down-regulation of MAVS-induced IFNβ production with supporting co-correlation of our findings using independent experimental assays. We further report that five of these same six identified SARS-CoV-2 genes also suppress MAVS-induced activation of type III IFNs (IFNλs), an associated but distinct set of anti-viral cytokines [22,39]. For five of the six genes, we share partial overlap of findings with that of others. Our identification of SARS-CoV-2 ORF7b as a suppressor of MAVS-induced IFN production is however novel. In sharp contrast to the suppression of IFN production, SARS-CoV-2 genes and

SARS-CoV-2 infected cells remain sensitive to activation by exogenous interferon, thus providing an opportunity for its use as a therapeutic intervention.

# Materials and methods

## Cell lines and virus infection

For culture experiments—human embryonic kidney 293T cells (HEK-293T) were maintained in Dulbecco's modified Eagle's medium (DMEM; Gibco 41965–039) supplemented with 10% fetal bovine serum (Gibco 12657–029), 1% pyruvate (Biological Industries 03-042-1B), and 1% penicillin–streptomycin (Biological Industries 03-031-1B). Vero cells (American Type Culture Collection [ATCC]) were maintained in Minimal Essential Media (MEM) supplemented with 10% heat-inactivated fetal bovine serum (FBS), 10 µM HEPES, 2 mM glutamine and antibiotics (100 units/mL Penicillin G, 100 µg/mL Streptomycin). For virus infection experiments, confirmed monolayers of Vero cells were infected with a known infectious dose of SARS-CoV-2 isolate hCoV-19/Australia/VIC01/2020 [40,41] in culture media, without the presence of FBS and supplemented with TPCK-Trypsin (Worthington) to a final concentration of 1µg/ml.

## Plasmids and transfection

Twenty nine cDNA recombinant plasmids (pLVX-EF1alpha) expressing 27 SARS-CoV-2 proteins, a mutant form of NSP5 and GFP vector control were generously provided by Nevan J. Krogan, as was reported previously [42]. pEF-BOS/MAVS, pEF-BOS/TRIF and IFNβ promotor-Firefly luciferase (pGL3/IFNβ1[-357>-55]) have been described elsewhere [43]. Other plasmids include pISRE-luciferase (Stratagene), pRL-TK (Promega- encoding Renilla Luciferase) and specifically for the RIG-I/MDA5 activation assays, pGL3-IFNβ [-280>20] was provided by Rongtuan Lin, McGill Universtiy, Qubec, Canada. GFP-CVS RABV (rabies virus CVS strain) P protein and N protein constructs were described elsewhere [44]. RIG-IN (CARD domains) was from Yoneyama et al [45]. MDA5 (CARD) was provided by Ashley Mansell, Hudson Institute of Medical Research, Australia. GFP-Mumps virus V protein-FLAG construct has been described elsewhere [46], using a plasmid from Curt Horvath. Cells were transfected using JetPrime (PolyPlus 114–07) for HEK-293T cells or using Lipofectamine 3000 (Thermo Fisher) for Vero cells, according to the manufacture's protocol. Opti-MEM medium (Gibco) was used for most transfections. In the case of RIG-I/MDA5 activation assays and ISRE promoter analysis transfections were performed using FuGENE HD (Promega) as described [47]. For the studies examining perturbations in cellular translation, the Krogan viral open reading frames (*EcoRI—ApaI fragments)* were transferred into an alternative vector (pMIGII), this which harbors an IRES-GFP following the viral gene, hence providing a means to observe transfection efficiency indirectly by fluorescence. These plasmids were then transfected into Vero cells. Successful transfection was assessed by observing cells for GFP expression.

## Luciferase activity assays

MAVS/TRIF induction: HEK-293T cells were transfected in 96 wells with IFNβ-Firefly luciferase (36 ng), Renilla luciferase (4 ng) with a combination of MAVS or TRIF (10 ng) and viral gene (50 ng) as indicated. 24 hours post transfection, cells were harvested and lysed with Glo Lysis Buffer (Promega, E2661) and Firefly/Renilla enzymatic activity was measured. Importantly, Renilla activity was measured with a 1:50 dilution of the original lysate, to ensure the signal was not over-saturated and within dynamic range.

RIG-I/MDA5 activation: HEK-293T cells were transfected in triplicate with pRL-TK (40 ng), pGL3-IFNβ (250 ng); MDA5-CARD, RIG-I-N or pUC-18 (125 ng); and plasmid to express the indicated viral protein or pUC-18 vector (250 ng). 24 hours post-transfection, cells were lysed for analysis by dual luciferase assay. The histograms show IFNβ-firefly luciferase, Renilla-luciferase and normalised luciferase (Firefly/Renilla) activity. Values given are mean ± Standard Error. To measure activity of the ISRE- containing promoter, HEK-293T cells were transfected in triplicate with pRL-TK (40 ng), pISRE-luc (250 ng) and plasmid to express the indicated viral protein or empty pUC-18 vector (250 ng) as has been described elsewhere ([47,48]. 24 hours post transfection, cells were treated for 8 hours with 1000 U/ml human IFNα (Hybrid/Universal PBL # 11200) before lysis and analysed by dual luciferase assay. The histogram shows normalised luciferase activity (Firefly/Renilla), mean ± SD.

## Quantitative PCR (qPCR) analysis

The expression levels of the different IFN-Is and IFN-I stimulated genes were measured using qPCR as described previously [49,50], with the following changes: For IFN-I transcript level analysis—MAVS and viral gene vectors were co-transfected into HEK-293T cells at a 1:5 ratio, respectively. mRNA was produced 24 hours post transfection. For IFN-I stimulated gene (ISG) transcript level analysis in culture, viral genes were transfected into HEK-293T cells. 24 hours post transfection cells were treated with 10 or 100 pM IFNβ for a further 24 hours before harvest. For quantification of IFN-I stimulated gene transcript levels after SARS-CoV-2 infection, Vero cells were infected with SARS-CoV-2 for 8 hours and treated with 1000 U/ml of human IFNα2 (Sapphire Bioscience) for an additional 16 hours. 24 hours post infection cells were harvested and cDNA was generated from total RNA. Gene expression levels were normalized against the housekeeping genes GADPH for Vero infected cells, or HPRT1 for HEK-293T cells. Primer sets for all qPCR experiments are listed in S1 Table.

## Western blots

For quantification of secreted IFNβ–HEK-293T cells were transfected with MAVS and viral gene vectors (ratio of 1:5 respectively). 24–48 hours after transfection cells were starved overnight using serum-free medium (DMEM, Gibco 41965–039), containing 1% non-essential amino acids (MEM Non-Essential Amino Acids Solution, Biological Industries 01-340-1B). Medium containing secreted proteins was collected and incubated with 0.02% Na Deoxycholate (DOC) for 15 minutes at RT. Secreted proteins were precipitated using Trichloroacetic acid (TCA) at a final concentration of 10% for 1 hour at RT. Pellets were washed twice with ice-cold acetone for 15 minutes at 4˚C. Acetone was completely removed and air-dried. The pellets were then dissolved in RIPA buffer (25mM TRIS, pH 7.4, 150mM NaCl, 1% NP-40, 0.5% DOC, 0.1% SDS). This experimental system was validated by running SDS-PAGE samples harboring increasing concentrations of purified IFNβ (50, 100, 200 & 400 nM; each in 30 ul volume) prior to WB analysis. Here we observed a band of increasing intensity at the expected mobility of ~22 KDa. Densitometry analysis demonstrated a linear relationship between IFN concentration and Unit intensity.

p-eIF2α, puromycin and p-STAT1 analyses: Vero cells were infected with SARS-CoV-2 for 24 and 36 hours, treated with 10 μg/ml puromycin (Life Technologies) or 1000 U/ml human IFNα (Sapphire Bioscience), for 20 and 30 minutes respectively, prior to harvesting. Soluble protein production, western blot preparation and analysis were performed as described earlier [50,51]. The following antibodies were used: mouse monoclonal anti-human IFNβ antibody (BioLegend Cat # 514002), rabbit anti-GFP (Abcam, Cat #ab290); rabbit anti-eIF2α (Invitrogen, Cat # 9721S); mouse anti-STAT1α (p91; Invitrogen Cat # AHO0832); rabbit anti-

phospho-STAT1 (Cell Signaling Technology Cat # 9167S); rabbit anti-actin (Sigma-Aldrich Cat # A2066), rabbit anti SARS-CoV-2 -N was used as described previously [52]. For detection of Strep-tagged SARS-CoV-2 recombinant gene products, we used mouse StrepMAB-Classic antibody (IBA Lifesciences, Cat # 2-15070-001).

## Immunofluorescence microscopy

Vero cells were infected with SARS-CoV-2 for 24 and 36 hours. Alternately, cells for immuno-fluorescence were transfected with the indicated viral genes for 24 hours. Puromycin (Life Technologies) was added to cells at a concentration of 10 µg/ml for 20 minutes prior to cell lysis. Cells were rinsed twice with phosphate buffered saline (PBS) and fixed with 4% v/v para-formaldehyde (PFA)/PBS for 15 minutes at RT. Fixative was removed and cells were permeabilized with 0.1% v/v Triton X-100 for 10 minutes at RT. Cells were rinsed twice with PBS and quenched with 0.2 M glycine for 10 minutes at RT. Cells were then rinsed with PBS and coverslips were incubated in primary antibodies diluted in 25 µl of 1% bovine serum albumin (BSA)/PBS for 1 hour at RT. Following incubation with primary antibodies, cells were washed thrice with 0.1% BSA/PBS. Coverslips were incubated in secondary antibodies diluted in 25 µl of 1% BSA/PBS for 45 minutes at RT. Cells were washed twice with PBS and incubated for 5 minutes with 4,6-diamidino-2-phenylindole (DAPI) (0.33 µg/ml) in PBS. Coverslips were rinsed twice with PBS and MilliQ water and mounted on cover-slides with ProLong Diamond (Life Technologies). Cells were analyzed using the Zeiss LSM710 confocal microscope.

Antibodies used: mouse anti-puromycin (Sigma-Merck, Cat # MABE343); Rabbit anti-STAT1 (Invitrogen Cat # PA5-81911); rabbit anti-phospho-STAT1 (Cell Signaling Technology Cat # 9167S); mouse anti-dsRNA—clone rJ2 (Sigma-Merck, Cat # MABE1134) and Alexa Fluor-488 and -594 conjugated species-specific IgG were purchased from Life Technologies.

## FACS

HEK-293T cells were transfected with viral gene vectors, each which encode in-frame 2x Strep tag fusion as described in [42]. Twenty-four hours post transfection, cells were collected and washed with PBS containing 0.5% BSA, before fixation with 2% PFA in PBS for 15 minutes at RT. After this, cells were washed twice, then permeabilized using ice cold methanol for 30 minutes at -20˚C. Cells were then washed before incubation with anti-Strep antibody for 1h on ice, according to the manufacture's protocol (IBA, strepMAB-Classic 2-1507-001), washed and incubated with biotinylated anti-mouse antibody for 1 hour on ice. Finally, cells were washed and incubated with Alexa Fluor 647 streptavidin (Jackson Immuno Research, AB_2341101) for 15 minutes on ice and were subjected to FACS analysis.

## IFN treatment of virus-infected cells

Vero cells were cultured in 24 well plates overnight. Upon confirmation of a >90% confluent monolayer, cells were washed once with serum-free culture media, then 0.5 mL inoculum containing 0.01 MOI (multiplicity of infection) of SARS-CoV-2 and 2µg/mL TPCK-trypsin were added, and cells returned to the incubator for 6h. 1000 IU of either IFNα2 (Sapphire Bioscience) or IFNβ (Merck-Serono) was then added in an equal volume to the inoculum (final concentration 500 IU) to n = 4–6 wells. Serum-free media was added to n = 4–6 wells as a no-treatment control. After a further 18h incubation, supernatants were harvested and assessed for presence of infectious virus by immediately performing a 50% tissue culture infectious dose assay (TCID$_{50}$). Briefly, samples were serially diluted in serum-free culture media and known volumes added to Vero cells in quadruplicate in the presence of 1µg/mL TPCK-trypsin. Presence of cytopathic effect was microscopically examined 3d later and the TCID$_{50}$/mL

determined via the method of Reed and Muench [53]. Independent experiments were performed—twice for IFNα treatment, and three times for IFNβ treatment, the data for each IFN-subtype which was then pooled for analysis. Statistical analysis was performed using the students' unpaired t-test method (GraphPad PRISM 9.0).

## Results

### Type I IFN treatment diminishes infectious virus titer of SARS-CoV-2

We first tested if added IFN-Is have the capacity to diminish SARS-CoV-2 infection. To do this, Vero cells were infected at a MOI of 0.01 and after 6 hours were treated with 500 IU of either IFNα or IFNβ and incubated for an additional 18 hours before determining viral titer by TCID50 assay. Both type I IFN variants induced a sharp reduction (>10-fold) in viral titer even after this short incubation time (Fig 1). These results demonstrate that, *added* IFN-Is have the capacity to diminish the advance of SARS-CoV-2 infection. This is an important and fundamental finding regarding the biology of SARS-CoV-2 in response to added interferons and may have therapeutic implications. However, this assay does not provide crucial insight into the dynamics of the natural IFN-I response in an infected host. Namely, 1) does early stage SARS-CoV-2 viral infection trigger the natural production of IFN-Is in a host, and 2) the observed decrease in viral titer to added IFN may be because of direct anti-viral effects taking place within an infected host cell. Alternatively (or additionally), added IFNs may be triggering the anti-viral activity in uninfected cells adjacent to the infected cell, thus perturbing advance of viral spread. For the remainder of this study we focus upon determining if/which SARS-CoV-2 genes have the capacity to inhibit IFN-I production. We also examined if/which of these viral genes have the capacity to block the innate antiviral response. These questions were addressed studying IFN-activation and response in both virus-infected cells as well as by studying individual SARS-CoV-2 genes transfected to mammalian cells.

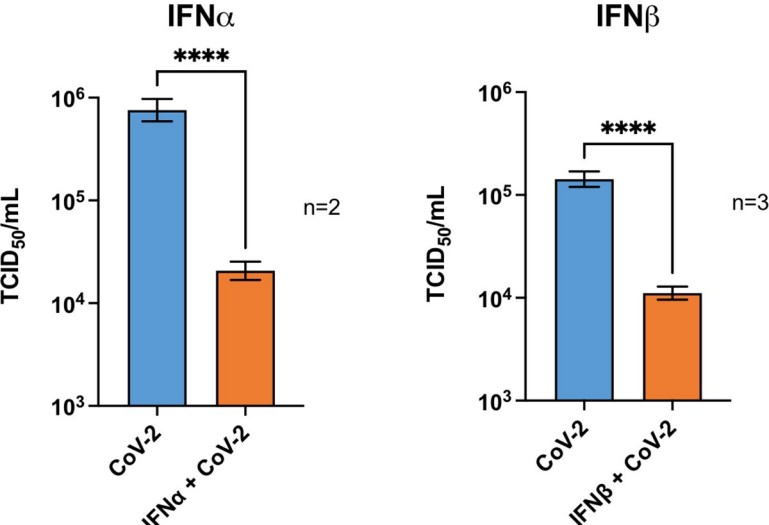

**Fig 1. SARS-CoV-2 is responsive to Type-I-IFN anti-viral activity.** Vero cells were inoculated at a MOI of 0.01 and after 6-hours were treated with 1000 IU of either IFNα, IFNβ, or an equal volume of media alone. 18 hours later, supernatants were harvested for $TCID_{50}$ analysis. N = number of repeat experiments performed. In each experiment, 4–6 independent samples per treatment type was measured. Bars represent mean +/- SEM; **** = p ≤ 0.0001 (Unpaired T-test).

## Similar expression of individual SARS-CoV-2 genes upon transfection and viral infection

Excessive gene over-expression may be of concern in transfection studies. Therefore, at the onset of this study we sought to quantify the degree of expression of transient transfected SARS-CoV-2 viral genes in comparison to that of cells infected with virus. Viral infection was quantified from a transcriptomic RNASeq study of Calu3 and Vero E6 cells, the primary data which was previously reported [54]. Cells were infected for 7 and 5 hours respectively, correct to a multiplicity of infection (MOI) of 20% for both cell types at time of harvest [54]. Viral transcript levels were determined in relation to that of the housekeeping gene HPRT1 (S1 Fig). Besides the determination of expression for individual viral genes, SARS-CoV-2 polygenic transcripts ORF1a and ORF1b, encoding NSP1-NSP11 and NSP1-NSP16 respectively were used to estimate the expression of NSP genes (S1A Fig). NSP genes were expressed at levels 30 to 60-fold higher than HPRT1 (S1B Fig). This is comparable to that of transfected individual SARS-CoV-2 genes transfected into HEK-293T cells, as determined by qPCR, where the NSP genes were expressed 14–36 fold higher than HPRT1. Levels of the other viral genes in infected cells fluctuated somewhat, depending on the cell type, but were similar to or higher than those for expression of the same genes by transfection. Exceptionally, the Spike (S) construct was expressed at much lower levels in transfected cells in relation to Calu3 or Vero E6 infected cells (S1B Fig).

In addition to transcript levels we verified viral protein translation, exploiting detection of the proteins via their fused 2xStrep tag [42]. Translation was determined using western blot (S2A Fig) or FACS (S2B and S2C Fig). All of the SARS-CoV-2 proteins were expressed, albeit not necessarily correlating with that for transcript levels (S1 Fig and also S7A and S7B Fig). In summary, all recombinant viral genes were demonstrated to be expressed at both the transcript and protein levels.

## Six SARS-CoV-2 gene products strongly suppress MAVS-induced IFNβ-luciferase promoter activity

The RNA-sensing RIG-I like receptors include RIG-I, MDA5 and LGP2, which interact with the downstream MAVS adapter, triggering activation of IFN transcription in an IRF3-dependent manner [55]. Due to its ease and high efficiency of transfection, we used the HEK 293T cell line to test for perturbation in IFNβ promotor activation by the individual SARS-CoV-2 gene products. HEK cells were co-transfected with plasmids to express MAVS, IFNβ-Firefly luciferase, Renilla for data normalization (driven by a constitutive promoter; in this case CMV), together with a plasmid expressing one of 27 SARS-CoV-2 viral genes or GFP. To monitor the basal activity of luciferase we transfected the cells with IFNβ-Firefly and Renilla plasmids alone ('no trigger'). Firefly luminescence values indicated that MAVS induced a strong (~30-fold) up-regulation in IFNβ promoter activity ("no trigger" compared to the GFP positive control; Fig 2A–2C). We observed that MAVS-induced IFNβ promotor activity was suppressed by > 10-fold by 6 different SARS-CoV-2 proteins, NSP1, NSP5, NSP6, NSP15, ORF6 and ORF7b (black bars). NSP5 is a protease which functions in part to facilitate cleavage of the viral ORF1a and ORF1b polyproteins into constituent peptides. A NSP5 mutant (C145A) which inactivates this proteolytic activity [42] abrogated repression of firefly activity, demonstrating a requirement of the enzymatic activity for NSP5 to suppress IFNβ production. Two other viral proteins, NSP13 and 14 showed a more modest suppression of MAVS-induced activation of the IFNβ promoter (shown in grey, Fig 2A). Of the six proteins which strongly inhibited Firefly activation, three (NSP1, NSP5 and NSP15) also induced a >10-fold decrease in Renilla normalization signal (Fig 2B). Thus, after normalization (by Firefly/Renilla) only

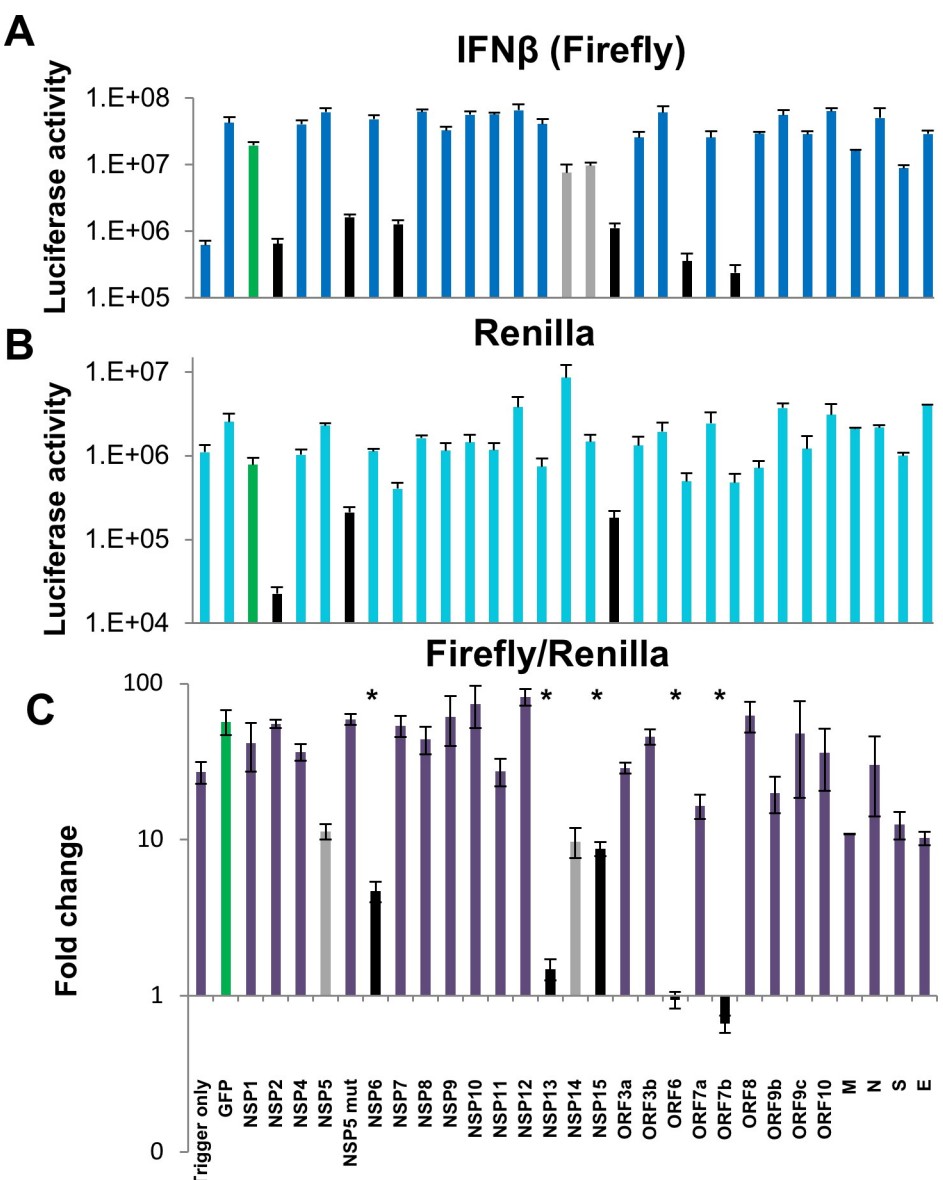

**Fig 2. Suppression of MAVS induced IFNβ-luciferase promotor activation by SARS-CoV-2 genes.** HEK-293T cells were co-transfected with IFNβ-firefly luciferase, Renilla luciferase, MAVS and a SARS-CoV-2 viral genes. 'No trigger' refers to cells transfected with Firefly and Renilla only (but no MAVS); 'Trigger only' refers to cells transfected with all components but without a viral gene. 24 hours post transfection luciferase activity was measured. IFNβ-firefly **(A)**, Renilla normalization signal **(B)** and normalized Firefly/Renilla luciferase **(C)** activities are presented separately. Data presented are means pooled from four independent experiments and their standard errors. Significance values are given in comparison to the GFP positive control (green bar): * P < 0.05. Black bars in the plot: 10-fold or greater repression of MAVS-induced promoter activity in comparison to the GFP control. Grey bars in histograms: Minor (< 10-fold) repression of MAVS-induced IFNβ activity.

NSP6, ORF6 and ORF7b remained strongly suppressive towards IFNβ promotor activity (>10-fold compared to GFP, Fig 2C). These findings are consistent with the possibility that NSP1, NSP5 and NSP15 effect non-specific perturbations of cellular activities. In addition to these genes, NSP13, which caused a rather modest decrease in IFNβ-firefly luciferase signal (Fig 2A), was associated with a 10-fold increase in Renilla signal (Fig 2B). As a result, the apparent suppression of IFNβ promoter activity by NSP13 is not real.

## SARS-CoV-2 ORF6 and ORF7b block both RIG-I and MDA5

MAVS is downstream to, and is activated by the RNA-sensing pattern recognition receptors RIG-I and MDA5. These sensors are non-identical in their RNA-binding properties and harbor non-redundant activities [2,56]. To test whether RIG-I, MDA5, or both are blocked by SARS-CoV-2, two of the SARS-CoV-2 genes (ORF6 and ORF7b) found to suppress MAVS-induced signaling were further evaluated for antagonistic function toward IFNβ-luciferase activation by transfection of HEK-293T cells with plasmids expressing the CARD domains of RIG-I or MDA5 or with an empty vector which served as a non-activation control. The CARD domains bind to MAVS following activation of RIG-I/MDA5 by RNA ligands, so expression of CARDs mimic activated RIG-I/MDA5. Mumps V (MuV V) protein, which antagonizes RIG-I, MDA5 and signals downstream of MAVS, was used as a positive control [12]. As with the MAVS-transfected cells, ORF6 and ORF7b, as well as MuV V protein, strongly suppressed (>10-fold) both RIG-I and MDA5-stimulated IFNβ promoter activity (Fig 3). Thus SARS-CoV-2 ORF6 and ORF7b gene products can block MAVS-dependent IFNβ production induced by both MDA5 and RIG-I signaling.

## TRIF induced IFNβ promotor activation is not suppressed by SARS-CoV-2 genes

Toll-like-receptor 3 (TLR3) is a pattern recognition receptor that is also activated by association with viral RNA PAMPs (Pattern Associated Molecular Patterns), specifically double-stranded RNA [57]. TLR3 promotes association with the adaptor molecule TRIF, which in turn triggers IFN-I production in an IRF3-dependent manner [58]. Indeed mice deficient in TRIF were found to be highly susceptible to SARS-CoV infection [59]. To investigate the role of SARS-CoV-2 genes in suppressing TLR/TRIF-induced IFNβ production we repeated the IFNβ promoter screen as was described for MAVS, but instead induced IFNβ production by TRIF [43]. TRIF transfection exerted a 100-fold up-regulation in IFNβ promotor activity in HEK-293T cells, as demonstrated by the Renilla-normalized luciferase signal for the GFP control vector (S3C Fig, green bar). However, unlike the findings with MAVS, none of the viral genes exerted a significant decrease in TRIF-induced IFNβ promoter activity. Once again, we observed a non-specific decrease in luciferase activity with NSP1 co-transfection, as both the raw IFNβ-Firefly signal and raw NSP1 Renilla signal were similarly suppressed (S3A and S3B Figs). We conclude that the SARS-CoV-2 viral proteins do not block TRIF-mediated IFNβ production.

## A common set of SARS-CoV-2 gene products suppress up-regulation of MAVS-induced IFNβ and IFNλ mRNA transcripts

The IFNβ-luciferase construct is artificial, and thus may not fully reflect transcription initiation from the IFNβ promoter. Furthermore IFN transcript levels may be subject to post-transcriptional regulation, such as alterations in mRNA stability, as was reported by others [60]. Thus, we chose to measure IFN-I transcript levels by qPCR. HEK-293T cells were transfected with MAVS along with individual SARS-CoV-2 genes for 24 hours prior to RNA extraction and expression level determination by qPCR. IRES-conjugated puromycin expression was monitored to demonstrate robust expression of each viral gene (S1 and S7A Figs). Measuring changes in critical threshold (CT) values for IFNβ normalized against the housekeeping gene HPRT1 (ΔCT), we found sharp reduction (>10-fold) in expression of MAVS-induced IFNβ gene by NSP1, NSP5, NSP6, NSP15, ORF6 and ORF7b (Fig 4A). These are the same genes identified by the non-normalized IFNβ-luciferase promoter activation assay (Fig 2A). As in

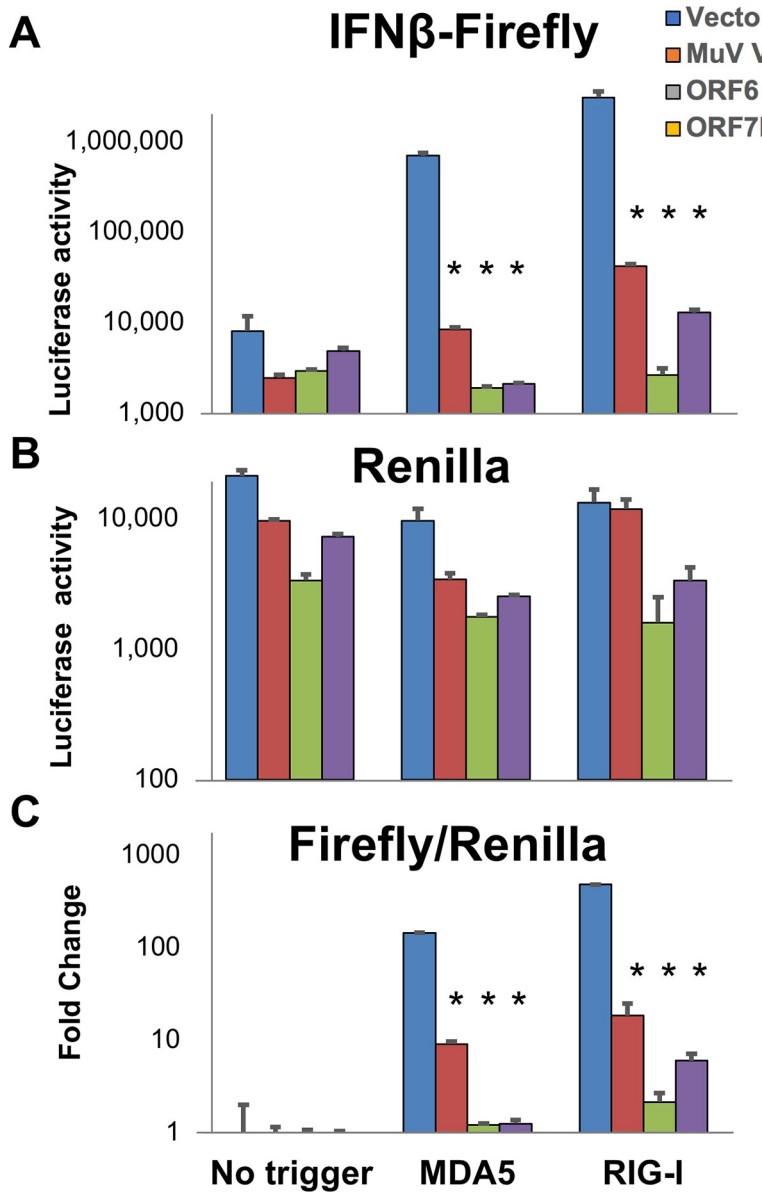

**Fig 3. Suppression of MDA5 / RIG-I induced IFNβ-luciferase promotor activation by SARS-CoV-2 genes.** HEK-293T cells were transfected with pRL-TK (Renilla normalization control vector), pGL3-IFNβ (IFNβ-luciferase vector) together with MDA5-CARD, RIG-I-CARD or pUC-18 (empty vector control) to assess the role of MDA5 and RIG-I in MAVs-dependent activation of IFNβ promoter activity. 24 hours post transfection luciferase activity was measured. Data presented are representative of three repeat experiments with IFNβ-firefly (**A**), Renilla-luciferase (**B**) and normalized Firefly/Renilla luciferase activity (**C**). Significance values are given in comparison to the empty vector control: * P ≤ 0.0001.

our previous assay, the NSP5 (C145A) mutant, which lacks proteolytic activity, failed to suppress activation of IFNβ mRNA transcript. Hence the qPCR results here are consistent with those found in quantification of pre-normalized IFNβ-luciferase signal, but not after Renilla normalization.

We next tested for perturbation of expression for cytokines of the IFNλ family. In a manner similar to IFNβ, these genes were MAVS-induced, as indicated by a sharp drop in ΔCT values for the MAVS + GFP vector control in relation to non-induced cells. IFNλ1 and IFNλ2/3

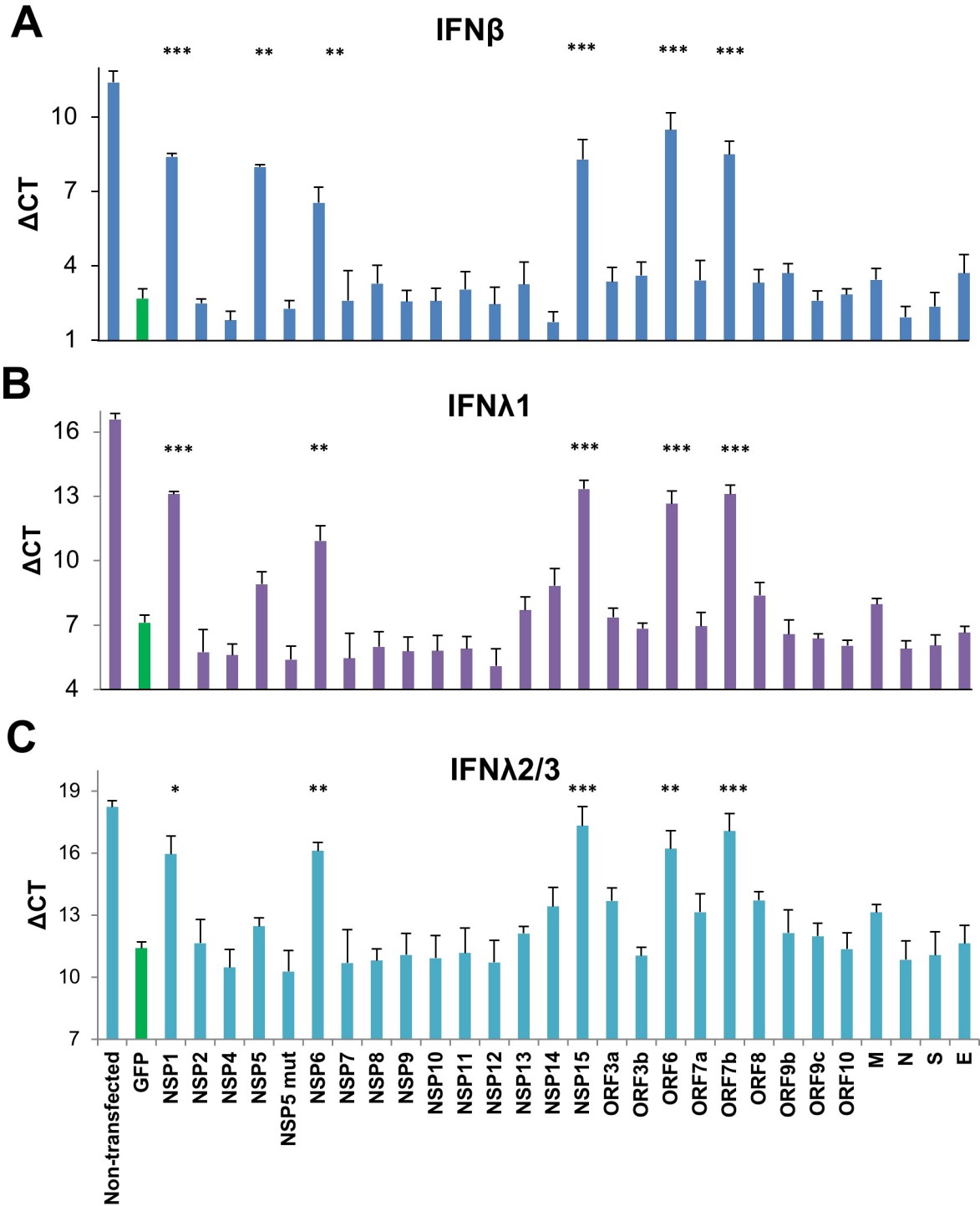

**Fig 4. Suppression of MAVS induced IFNβ, IFNλ1 and IFNλ2/3 gene expression by SARS-CoV-2 genes determined by qPCR.** HEK-293T cells were transfected with MAVS and a SARS-CoV-2 viral gene or GFP. 24 hours post transfection, transcript levels were analyzed by qPCR. The data presented are expression levels normalized to housekeeping gene HPRT1 (ΔCT). Results are shown for **(A)** IFNβ; **(B)** IFNλ1; and **(C)** IFNλ2/3 (common primers amplify both IFNλ2 and IFNλ3). Data presented are means of 2–4 independent experiments ± SEM. Significance was compared to the GFP positive control: * P < 0.05, ** P < 0.001, *** P < 0.0001.

(common primers amplify both IFNλ2 and IFNλ3) were suppressed by five of the same six SARS-CoV-2 genes found to suppress IFNβ transcript (Fig 4B and 4C). It should be noted that at least for this cell type, there was a much greater abundance of MAVS-induced IFNβ transcript than of IFNλ1 or IFNλ2/3 transcript (CT values for GFP control: 2.7, 7.1 and 11.4, respectively), indicating that IFNβ is expressed at least 10-fold higher than IFNλ1, 2 or 3. To obtain a more complete picture on MAVS induced IFN expression, changes in mRNA for several IFNα genes (IFNα2, IFNα4, IFNα6, and IFNα10) as well as type II IFNγ were also evaluated (S4 Fig). In contrast to IFNβ and IFNλ, we did not detect MAVS-induced up-regulation of expression for any of these genes, nor their subsequent suppression by SARS-CoV-2 genes (S4 Fig).

## The same SARS-CoV-2 gene products suppress MAVS-induced IFNβ secretion

Both the IFNβ-luciferase and qPCR assays do not assess the amount of IFNβ protein that is secreted following MAVS activation or its possible inhibition by SARS-CoV-2 proteins. For this, HEK-293T cells were transfected with plasmids to express MAVS and a selection of SARS-CoV-2 viral genes for 48 hours. In addition to the six viral genes identified by both the IFNβ-luciferase and qPCR assays, we also evaluated ORF3b and NSP14, which were previously reported by others to block IFNβ promoter activity [27,30], along with ORF8 serving as negative control. To evaluate IFNβ expression and secretion, the cell media were collected, and proteins were concentrated by TCA precipitation before analysis by western blot for IFNβ (see Materials and Methods). All of the six SARS-CoV-2 viral genes identified as inhibitory in IFNβ-luciferase and qPCR assays (NSP1, NSP5, NSP6, NSP15, ORF6 and ORF7b) were associated with a strong decrease (>10-fold) in IFNβ secretion (Fig 5A and 5B). Additionally, we detected a weak/moderate decrease in IFNβ secretion for NSP13 and NSP14, as demonstrated by quantification of the band intensities (Fig 5B), which is in line with the reduced IFNβ-luciferase expression (non-normalized; Fig 2A).

## SARS-CoV-2 has no significant suppressive effect on ISG activation

Several publications have suggested that some SARS-CoV-2 genes block IFN induced signaling in addition to IFN production (see introduction). To test for this, we transfected HEK-293T cells with plasmids encoding individual SARS-CoV-2 genes or control GFP for 24 hours, followed by stimulation with 100 pM IFNβ for another 24 hours. mRNA was then extracted and ISG expression evaluated by qPCR. IRES-conjugated puromycin expression was monitored to demonstrate robust expression of each viral gene (S7B Fig). Expression of four representative ISGs—MX1, MX2, OAS2 and CXCL10 were evaluated (Fig 6). A sharp increase in expression (lower ΔCT values) of these genes was observed upon IFNβ stimulation, using cells expressing GFP as a control, in relation to un-treated cells. None of the 27 SARS-CoV-2 genes altered the elevated expression of these ISGs (Fig 6).

To account for saturation of the system by this high concentration of IFNβ (100 pM), which may mask an inhibitory effect on the ISGs by viral genes, we repeated this experiment with 10 pM IFNβ (S5 Fig). We observed here weak expression of MX2 and OAS2 with IFNβ treatment. Once again, none of the transfected SARS-CoV-2 genes blocked expression of these ISGs (S5 Fig), supporting that SARS-CoV-2 does not directly act to suppress ISG production.

To further validate these findings by an alternative method, ISRE promoter activity was evaluated using a luciferase reporter assay (Fig 7) for a subset of SARS-CoV-2 genes, ORF3a, ORF3b, N and ORF6 after treatment with IFNα for 16 hours. The rabies virus phospho-protein (RABV P1) served as a positive control for suppression of promoter activity, as it blocks

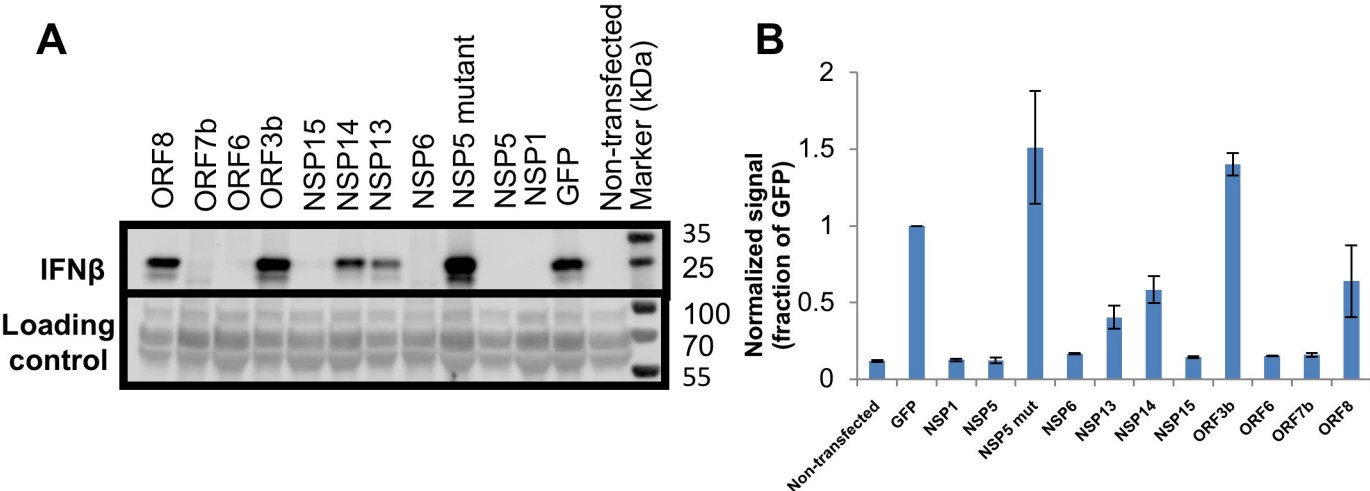

**Fig 5. Suppression of MAVS induced IFNβ protein secretion by SARS-COV-2 genes. (A).** Upper panel: HEK-293T cells were transfected with MAVS and a viral gene. 48 hours post transfection secreted proteins were collected and analyzed by western blot, using an antibody specific to human IFNβ. Lower panel: The transfer membrane was stained with Ponceau S, this which served as a sample loading control. **(B).** Quantitative analysis of (A) (average of two independent experiments ± SEM).

IFN-I signaling by preventing STAT1 nuclear localization [11]. As anticipated, none of the SARS-CoV-2 proteins blocked IFN-induced ISRE activation, in sharp contrast to that of the RABP P1 positive control.

## SARS-CoV-2 infection is associated with decreased IFN-induced STAT1, yet ISGs remain activated

IFN-I induces signaling through the phosphorylation of STAT1 and STAT2, which together with IRF9 translocate to the nucleus, where they serve as transcription factors required for downstream signaling [50]. To determine whether viral infection might alter STAT1 activation, Vero cells were infected with SARS-CoV-2 for 24 hours, prior to the addition of IFNα2. Immunocytochemistry staining of individual cells was used to follow STAT1 (Fig 8A and 8B) and pSTAT1 (Fig 8C and 8D) levels. As expected, mock-infected cells were observed to have predominantly cytoplasm-restricted STAT1 and undetectable levels of pSTAT1 in the absence of IFNα2 (Fig 8A i-iii and 8C i-iii). Intriguingly, this was also true for cells infected with SARS-CoV-2 (Fig 8A iv-vi and 8C iv-vi), suggesting that infection does not induce phosphorylation nor nuclear translocation of STAT1. IFN addition to mock-infected cells led to a robust increase in nuclear translocation of both total and phosphorylated STAT1 (Fig 8A vii-ix and 8Cv ii-ix). However, IFNα2 addition to SARS-CoV-2 infected cells was associated with substantial attenuation of both phosphorylation and nuclear translocation of STAT1 (Fig 8A x-xii) and pSTAT1 (Fig 8C x-xii). Quantification of replicate experiments are presented for STAT1 and pSTAT1 (Fig 8B and 8D, respectively). Virus-induced decrease in total and phosphorylated STAT1 levels was confirmed by western blot, in cells infected with SARS-CoV-2 for 24 and 36 hours in the presence or absence of exogenously added IFNα2 for 30 minutes before harvesting the lysates (Fig 8E).

A decrease in STAT1 activation at face value would predict a corresponding decrease in IFN-induced ISG activation. However, our findings were contrary to this assumption. Vero cells were infected with SARS-CoV-2 for 8 hours prior to the addition of IFNα2 for an additional 16 hours, after which the cells were lysed and RNA was extracted for qPCR analysis.

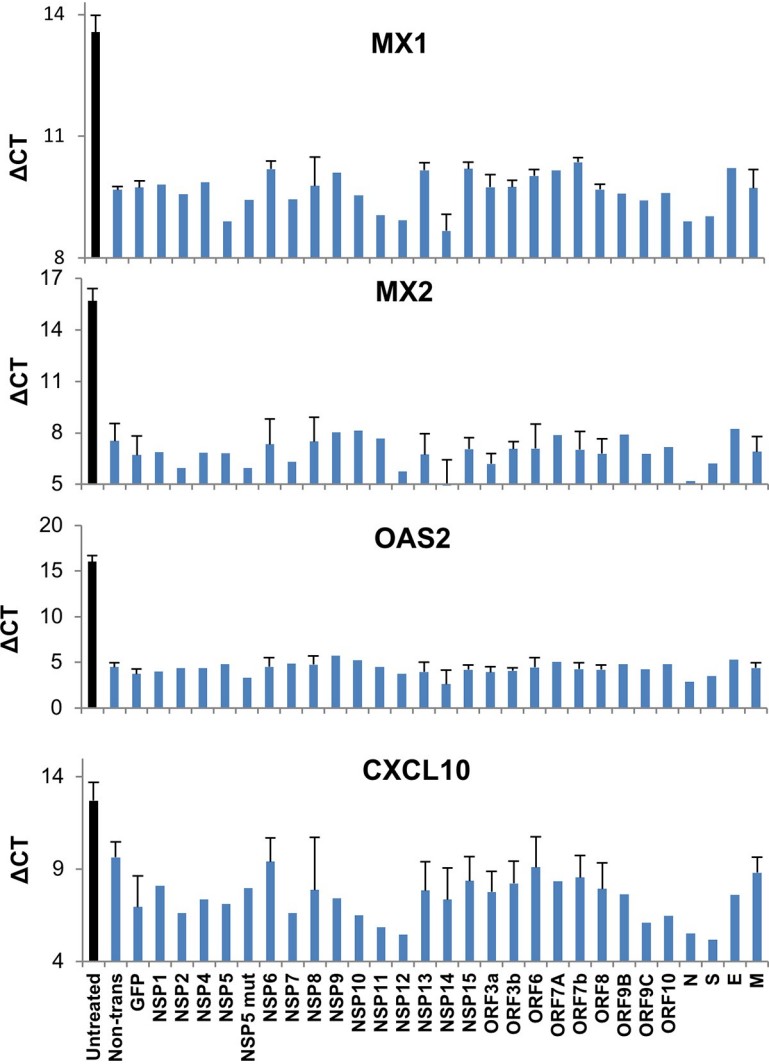

**Fig 6. SARS-CoV-2 genes fail to suppress activation of Interferon-Stimulated Genes (ISGs).** HEK-293T cells were transfected with a SARS-CoV-2 viral gene, or GFP control. 24 hours post transfection cells were treated with 100 pM IFNβ for an additional 24 hours before harvest. Transcript levels of four ISGs were analyzed by qPCR. The data presented are expression levels normalized to housekeeping gene HPRT1 (ΔCT). Average values ± SEM were calculated based on 2–3 independent experiments.

Here we tested the anti-viral ISGs MX1, MX2, OAS2 and IFIT1. All ISGs demonstrated a strong activation of expression in the presence of IFNα, regardless of whether the cells were infected with virus or not (Fig 8F). We also measured SARS-CoV-2 E transcript levels as a proxy-indicator of virus infection levels. Whereas a strong activation of CoV-2 E transcript was found for the virus-infected culture, this was decreased 8-fold in the presence of IFNα (ΔCT of -9.55 vs. -6.79 respectively, Fig 8F). This finding supports that in infected cells IFNα both activates ISGs and also exerts antiviral SARS-CoV-2 activity.

To understand why the reduced level of pSTAT1 had no effect on IFN-I induced gene activation, we depleted STAT1 using siRNA, this time using HeLa cells. While the level of total STAT1 was reduced by >5-fold, the effect on pSTAT1 levels 30 minutes and 6 hours after IFNα2 administration was small (S6A and S6B Fig). Furthermore, the residual STAT1 protein was sufficient to induce the same level of IFN-I induced ISG expression, as well as activation of

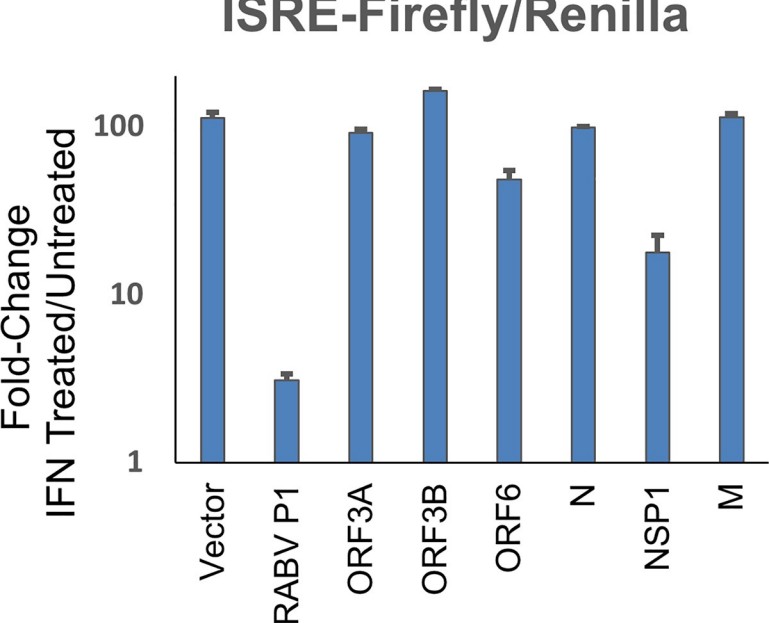

**Fig 7. ISRE-luciferase promotor activation inhibition by SARS-CoV-2 genes.** HEK-293T cells were transfected with pRL-TK (Renilla normalization control vector), pISRE-luciferase and viral plasmids to express the indicated viral protein or empty pUC-18 vector. The rabies virus phospho-protein–RABVP1 served as positive control. 24 hours post transfection cells were treated for 8 hours with 1000U/ml IFNα before cells were lysed and analysed by dual luciferase assay. Data shown are of the ratio of IFN-treated / untreated cells calculated from normalised luciferase activity (Firefly/Renilla) for each given viral gene. The data (average values ± SEM) is representative of three independent experiments.

an anti-viral response as the control (S6C and S6D Fig). Only complete knockout of STAT1 resulted in a large reduction of IFN-I induced ISG expression and a reduction in IFN-induced antiviral activity [50,61]. Our results support that only a small pool of STAT1 is required for IFN-Is to induce robust activation of ISGs and antiviral activity.

## SARS-CoV-2 infection results in a shutdown of protein translation via eIF2α

Viruses often modulate and/or manipulate the host translational machinery to induce a bias towards viral protein production over the host. For SARS-CoV, NSP1 was found to drive this activity [62]. If a similar mechanism is present in SARS-CoV-2 NSP1, it would provide a rational explanation to its ability to block IFNβ production. The phosphorylation of eIF2α impedes host cell translation, and can be induced as an antiviral mechanism through activation of its kinase PKR by viral RNA PAMPs (dsRNA); detection of phosphorylated eIF2α (p-eIF2α) is an indicator of this mechanism. Vero cells were either mock or SARS-CoV-2 infected for 24 and 36 hours, and whole cell lysates were harvested for western blotting. Over the course of SARS-CoV-2 infection we observed a gradual increase in eIF2α phosphorylation (Fig 9A). These results indicate that the infected cell is responding to viral infection and attempting to shut down protein translation via phosphorylation of eIF2α.

To quantify protein translation activity, measurement of incorporation of puromycin into newly synthesized polypeptide chains can be used as a proxy indicator [63]. Puromycin inhibits translation by incorporation to active ribosomes. Therefore, the amount of active ribosomes in a cell can be estimated using anti-puromycin antibodies. To determine the effect of

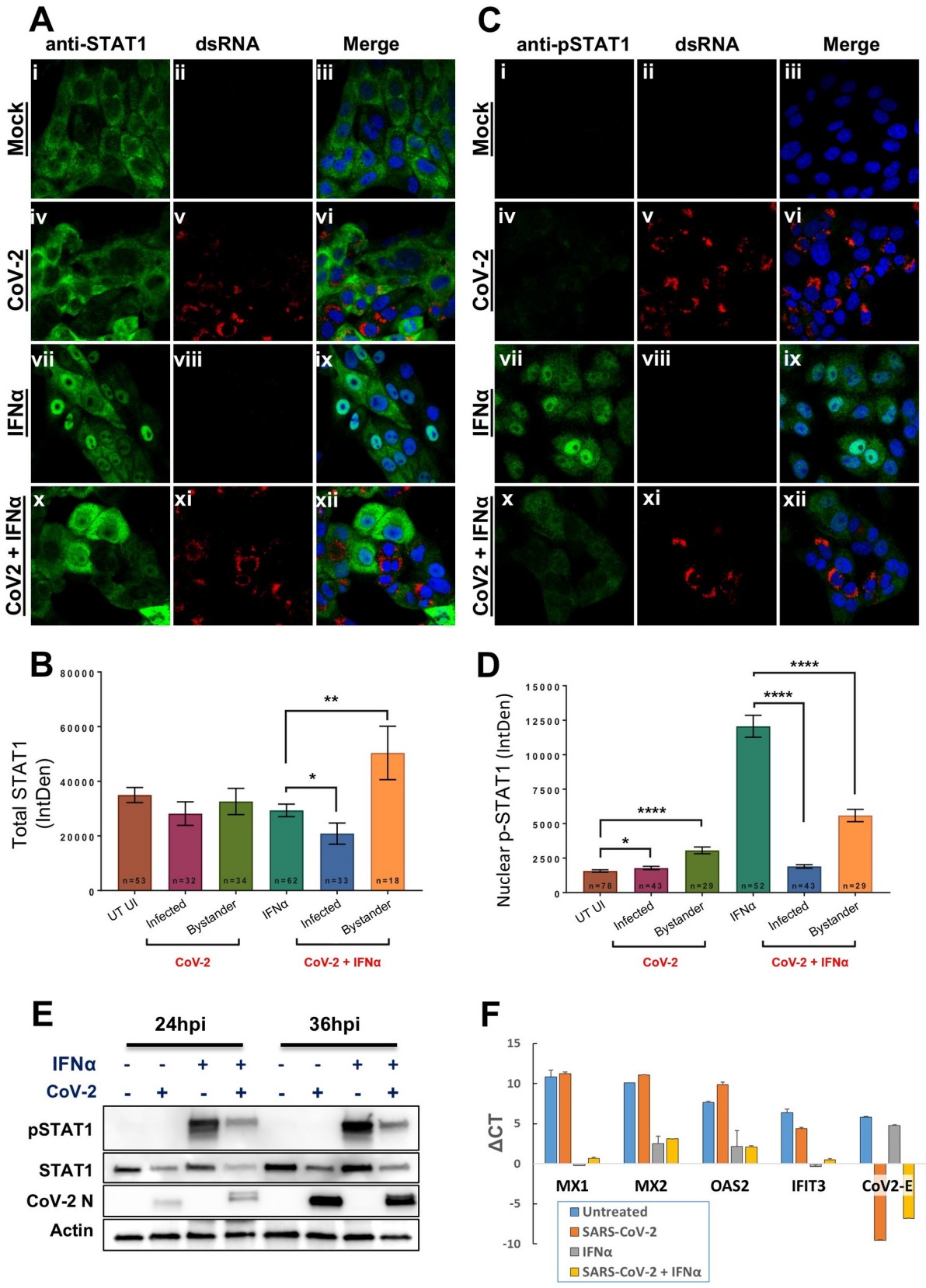

**Fig 8. SARS-CoV-2 infection reduces total STAT1 levels and subsequent phosphorylation and nuclear translocation of STAT1.** Vero cells were infected with SARS-CoV-2 for 24 hours before IFNα was exogenously added for 30 minutes prior to analysis. **(A)** Immunofluorescence (IF) analysis of total STAT1 (green in panels i, iv, vii and x) in mock- (panels i-iii and vii-ix) and CoV-2-infected cells (panels iv-vi and x-xii) detected with anti-dsRNA antibodies (red in panels v and xi) in the absence (panels i-vi) or presence (panels vii-xii) of IFNα. **(B)** Quantification of total STAT1 (whole cell) assessed by IF analysis. Number of cells used are indicated from replicate experiments. Average measurement of fluorescence +/- SEM is depicted. Significance was assessed by students *t*-test: * p<0.05 and ** p<0.01. **(C)** and **(D)** Same analysis as described above for the detection of pSTAT1, except the quantitation depicts the amount of p-STAT1 observed within the nucleus. Average measurement of fluorescence +/- SEM is depicted. Significance was assessed by students *t*-test: * p<0.05 and **** p<0.0001. **(E)** Western blot depicting the abundance of both STAT1 and pSTAT1 in mock- and infected cells after exogenous addition of IFNα for 30 minutes. SARS-CoV-2 Infected cells were detected using anti-N antibodies and anti-actin antibodies served as a loading control. **(F)** qPCR of SARS-CoV-2 infected Vero cells. The data presented are expression levels normalized to housekeeping gene GAPDH (ΔCT). Results presented are means of three repeat experiments +/- SEM.

increasing p-eIF2α on host translation, Vero cells were infected with SARS-CoV-2 for 24 or 36 hours, treated with puromycin for 20 minutes and cells were lysed for western blot analysis (Fig 9B), or fixed for immunofluorescence (IF) (Fig 9C). Western blot analysis of puromycin incorporation shows reduced amounts of puromycin in infected cells, which is both decreased by 36 hours post infection (h.p.i), when eIF2α phosphorylation peaks (Fig 9B). Immunofluorescence analysis (Fig 9C) further confirms this finding, showing large reduction in puromycin incorporation in infected cells (panels iv-vi and x-xii), with puromycin stained green (panels i, iv, vii and x) and dsRNA (infection marker) stained red (with an anti-dsRNA antibody in panels ii, vviii and xi). Quantification of the fluorescence intensity between these samples and replicate experiments is depicted in Fig 9D. Together, these results indicate that SARS-CoV-2 infection leads to the increased phosphorylation of eIF2α, and this increase in p-eIF2 α results in the shutoff of host cell translation, yet viral protein production still ensues. Interestingly, puromycin incorporation in cells adjacent to the infected cells (bystander cells) was elevated compared with cells in non-infected samples (Fig 9D). This suggests communication between infected and un-infected cells, with the increased translation possibly being a part of the innate host defense mechanism against viruses.

## SARS-CoV-2 NSP1 contributes to cellular translational shutdown

To evaluate which viral gene products were responsible for the translational arrest in SARS-CoV-2-infected cells, we co-expressed a viral gene (namely N, NSP1, NSP5, NSP5-C145A, ORF3a, ORF6, and ORF10) with GFP for detection and identification of transfected cells. The plasmids were transfected into Vero cells and we assessed the impact of their expression on cellular translation. After 24 hours, cells were treated with puromycin for 20 minutes prior to immunofluorescence analysis (Fig 10A). Of all the genes tested, NSP1 transfection alone was associated with an abrogation of puromycin staining, indicating a significant impact on host translation (Fig 10A vii-ix; marked with stars). This was further supported by western blot analysis of cells transfected with NSP1 (Fig 10B). Furthermore, NSP1 transfection was also associated with a decrease in IFN-induced pSTAT1 expression, although this was not found to associate with a decrease in total STAT1 levels (Fig 10C). Our findings are consistent with recent publications by Schubert et al and others [64–67], which demonstrated that NSP1 blocks cellular translation by binding to the human 40S subunit in ribosomal complexes to inhibit translation, but we further extrapolate that one of its downstream effector functions is to dampen IFN-production.

## SARS-CoV-2 Δ382 variant harbors a defective ORF7b

A SARS-CoV-2 variant harboring a 382 nucleotide deletion was discovered in Southeast Asia, this displaying relatively mild clinical symptoms [68]. The mutant had limited evolutionary

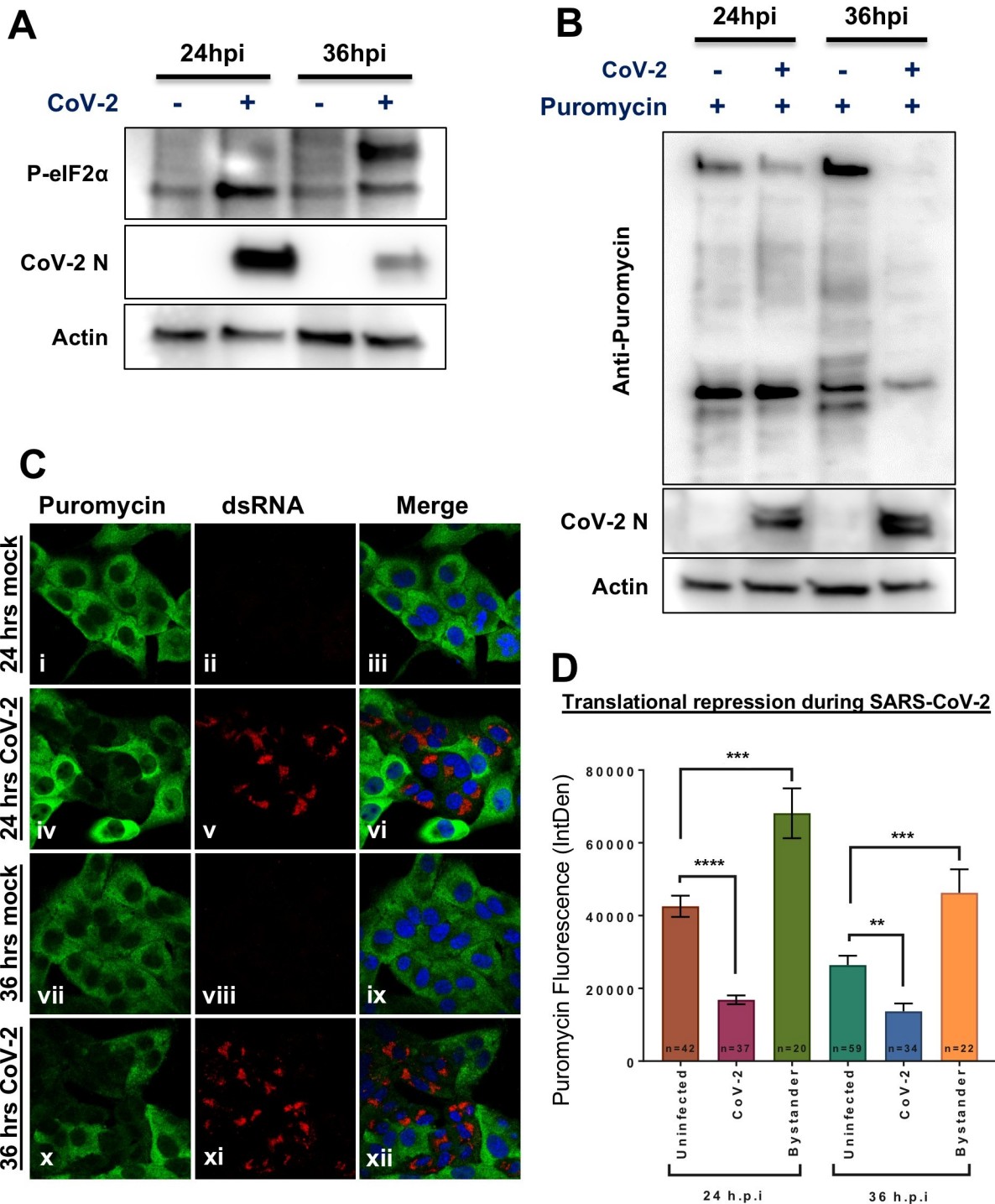

**Fig 9. SARS-CoV-2 induces phosphorylation of elF2α and the subsequent attenuation of host-cell protein translation.** Vero cells were infected with SARS-CoV-2 for 24 or 36 hours before puromycin was exogenously added for 20 minutes prior to analysis. **(A)** Western blot depicting the increased phosphorylation of elf2α over time (upper panel) Infected cells were detected using a SARS-CoV-2 anti-N antibodies (middle panel). Anti-actin antibodies (lower panel) served as a loading control. **(B)** Western blot demonstrates decreased puromycin incorporation over the course of infection: The incorporated puromycin was detected using anti-puromycin antibodies (upper panel). Infection and loading controls are shown, as was described in (A). **(C)** IF analysis of puromycin incorporation as detected using anti-puromycin antibodies and AF488 (green in panels i, iv, vii and x) in mock- (panels i-iii and vii-ix) and CoV-2-infected cells (panels iv-vi and x-xii) were detected using anti-dsRNA antibodies counterstained with AF594 (red in panels v and xi). **(D)** Quantification of puromycin incorporation in mock-infected, SARS-CoV-2-infected and bystander cells at 24 and 36 h.p.i (hours post infection). Average measurement of fluorescence +/- SEM is depicted. Significance was assessed by students $t$-test: * $p < 0.05$, ** $p < 0.01$, *** $p < 0.001$ and *** $p < 0.0001$.

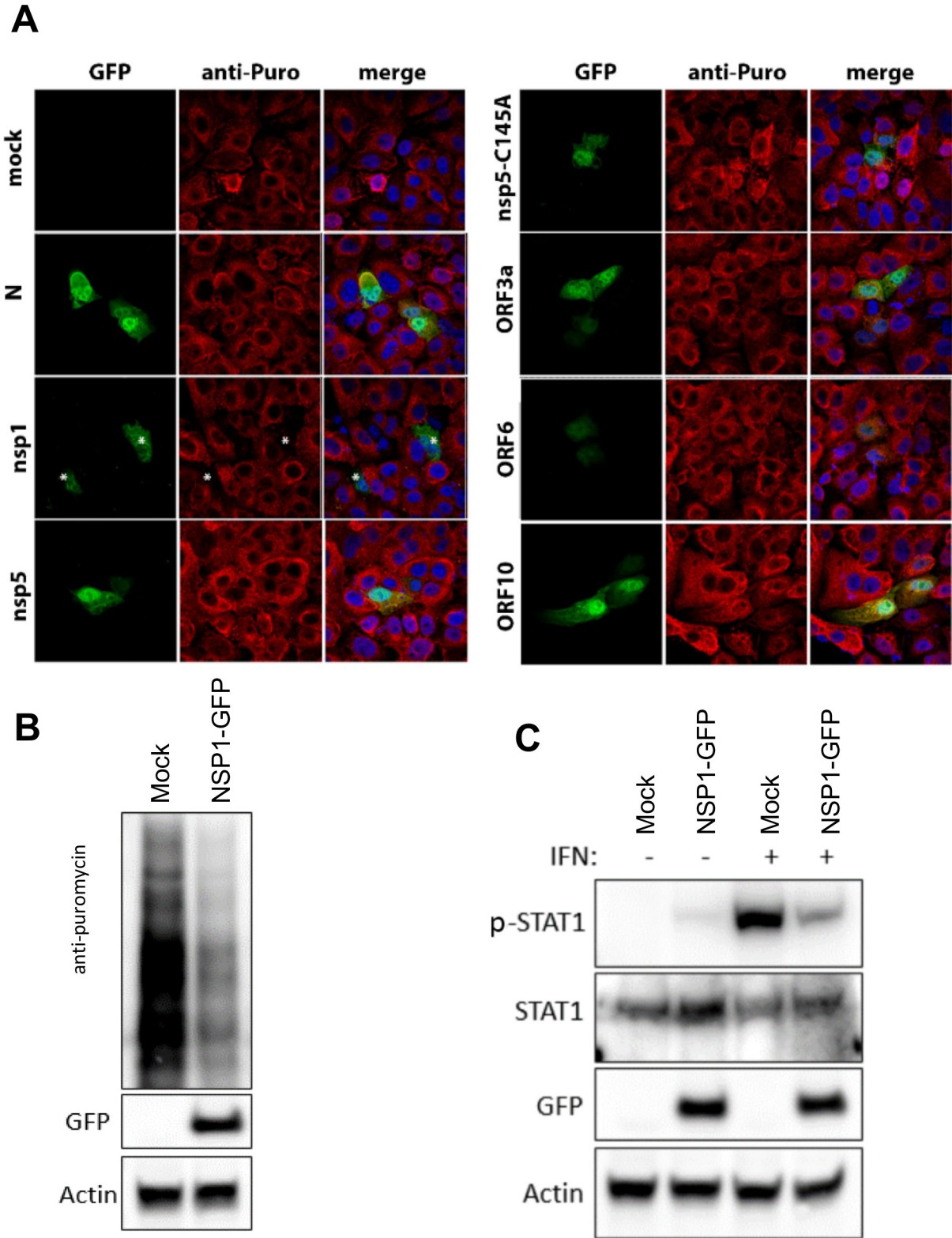

**Fig 10. SARS-CoV-2 NSP1 attenuates host cell protein translation restricting the abundance of STAT1. (A)** Vero cells were transfected with cDNA expression plasmids encoding vector only (panels i-iii), or the indicated SARS-CoV-2 protein co-expressed with GFP (panels i, iv, vii, x, xiii, xvi, xix, xxii); N (panels iv-vi), NSP1 (panels vii-ix), NSP5 (panels x-xii) NSP5-C145A (panels xiii-xv), ORF3a (panels xvi-xviii), ORF6 (panels xix-xxi) and ORF10 (panels xxii-xxiv). 24 hours after transfection puromycin was exogenously added for 20 minutes prior to

detection by anti-puromycin antibodies and AF594 (panels ii, v, viii, xi, xiv, xvii, xx and xxiii) by IF analysis. Merged images are provided in panels iii, vi, ix, xii, xv, xviii, xxi and xxiv. Stars *—NSP1-transfected cells (green) demonstrate a sharp reduction in puromycin signal (red) **(B)** Western blot showing a decrease in puromycin incorporation during expression of SARS-CoV-2 NSP1 for 24 hours in Vero cells. The incorporated puromycin was detected using anti-puromycin antibodies. Transfected cells were detected using anti-GFP antibodies and anti-actin antibodies served as a loading control. **(C)** Western blot depicting the abundance of both STAT1 and pSTAT1 in mock- and NSP1-transfected Vero cells after exogenous addition of IFNα for 30 minutes. Transfected cells were detected using anti-GFP antibodies. Anti-actin antibodies served as loading control.

fitness, and was lost from the region over time. Although this variant is commonly described to ablate expression of ORF8 which was assumed to be the causative factor of the reduced fitness of this variant, the Δ**382** also extends into the 3' end of the ORF7b open reading frame, resulting in a frame-shift mutation immediately after the predicted transmembrane domain and replacement with five unrelated amino acids prior to protein termination [69] (Fig 11A). We generated this ORF7b Δ382 mutant gene, cloned it into pLVX-EF1alpha, and probed its ability to suppress MAVS-induced interferon production by transfection into HEK-293T cells. Whereas WT ORF7b resulted in strong suppression of MAVS-induced IFNβ, IFNλ1 and IFNλ2/3 expression in relation to mock-control (GFP) transfection, the ORF7b Δ382 variant demonstrated total loss of this suppressive activity (Fig 11B). Therefore, our results demonstrate that the mutated ORF7b present in SARS-CoV2 Δ382 has lost its ability to block MAVS-induced IFN-I production through ORF7b, which could contribute to the reduced pathogenicity described for this mutant virus strain.

## Discussion

In this study, we interrogated 27 SARS-CoV-2 encoded proteins (with the exception of NSP3 and NSP16) for their ability to suppress either IFN-I signaling or IFN-I, II or III production.

**A**

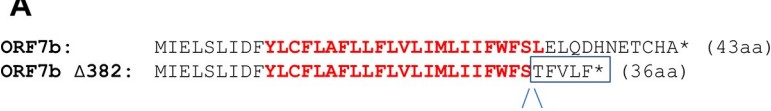

```
ORF7b:        MIELSLIDFYLCFLAFLLFLVLIMLIIFWFSLELQDHNETCHA* (43aa)
ORF7b Δ382:   MIELSLIDFYLCFLAFLLFLVLIMLIIFWFSTFVLF* (36aa)
                                             /\
                                           Δ382
```

**B**

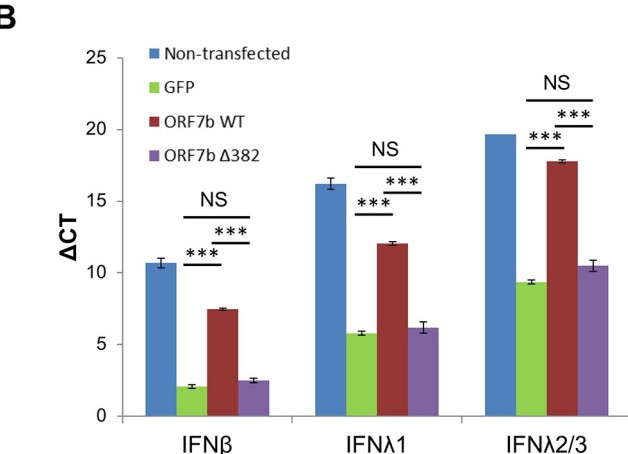

**Fig 11. SARS-CoV-2 Δ382 mutant does not suppress MAVS-induced IFN-production. (A)** Alteration of ORF7b protein sequence as a consequence of a 382bp deletion in SARS-CoV-2. The residues in red show the predicted transmembrane domain. The star represents a stop-termination codon. **(B)** qPCR was performed (as described in Fig 4) comparing MAVS-induced suppression of IFN mRNA levels by wild-type (WT) SARS-CoV-2 ORF7b and mutant ORF7b-Δ382. Significance was compared to the GFP positive control, *** $P < 0.0001$. NS—not significant.

**Table 1. Summary of experimental evidence emanating from this study, supporting inhibitory activity of individual SARS-CoV-2 genes in repressing type I and type III IFN production.**

|  | IFNβ-Luc | IFNβ-Luc / Renilla | IFNβ mRNA | IFNβ Protein | IFNλ1 mRNA | IFNλ2/3 mRNA |
|---|---|---|---|---|---|---|
| NSP1 | +++ | - | +++ | +++ | +++ | + |
| NSP5 | +++ | + | ++ | +++ | - | - |
| NSP6 | +++ | +++ | ++ | +++ | ++ | ++ |
| NSP13 | + | +++ | - | + | - | - |
| NSP14 | + | + | - | + | - | - |
| NSP15 | +++ | + | +++ | +++ | +++ | +++ |
| ORF6 | +++ | +++ | +++ | +++ | +++ | ++ |
| ORF7b | +++ | +++ | +++ | +++ | +++ | +++ |
| **Inhibition intensity** | | | | | | |
| **None** | | | | - | | |
| **Weak** | | | | + | | |
| **Moderate** | | | | ++ | | |
| **Strong** | | | | +++ | | |

Table 1 provides a summary of the results obtained from the three methods applied: IFNβ-luciferase reporter activity (with and without normalization using Renilla expression), probing mRNA levels by qPCR and direct detection of secreted IFNβ protein. The same 6 proteins (NSP1, NSP5, NSP6, NSP15, ORF6 and ORF7b) completely abrogated MAVS induced IFNβ production, when the non-normalized results for the IFNβ-luciferase reporter were used.

Regarding these assays, the direct measurement of IFNβ protein secretion is perhaps the most biologically-relevant one to investigate viral gene suppression of MAVS-induced IFNβ production (Fig 5). The identification of the same six genes (and additionally to a more minor degree the repression of NSP13 and NSP14) using this alternative methodology further solidifies our findings. While there is consensus that SARS-CoV-2 can block IFNβ production, the identification of a number of alternative SARS-CoV-2 viral genes that mediate this were described in different publications. To further, independently validate our results, a sample of the viral genes (ORF6 and ORF7b) were tested independently by co-authors from separate institutions (MAVS induction was performed at the Weizmann Institute, Israel while MDA5/RIG-I induction was performed at Monash University, Australia) providing complementary findings. With exception of a common source of SARS-CoV-2 expressing transgenes (kindly provided by Nevan Krogan, UCSF), all reagents and protocols were performed independently. Whereas it is always possible that recombinant constructs from different origins can provide different phenotypic outputs, perhaps due to differential expression levels or codon usage, we provide evidence that at least at the transcript level, the SARS-CoV-2 transgenes are not unduly overexpressed relative to the same gene products expressed in SARS-CoV-2 infected cells (S1 Fig).

In relation to IFN λ induction, our mRNA studies demonstrate that the five from the same six SARS-CoV-2 proteins responsible for repression of MAVS-induced IFNβ were also found to repress IFNλ1 as well as IFNλ2/3 (these two genes are nearly identical in sequence). Interestingly, we failed to detect either MAVS-induced activation of a number of IFNα genes as well as IFNγ, nor detect their subsequent repression by SARS-CoV-2 genes (S4 Fig). It has been reported elsewhere in animal virus challenge models however that MAVS can activate both IFNβ and IFNα [70]. It is possible that the differential ability of MAVS to activate IFNβ, IFNα or the both of them may be related to the cell-type in which the activation has taken place.

There is a notion that IFNα is preferentially activated in cells of the immune lineage (for example in plasmacytoid dendritic cells) and not directly by cells of the lung epithelia [71]. Severe COVID-19 is indeed associated with a sharp up-regulation of IFNα along with localized infiltration of dysregulated cells of the myeloid lineage, consistent with the notion that IFNα at this later stage of disease could be immune-cell derived [72,73]. Consistent with this notion, even though dsRNA triggers activation of TLR3, which signals through TRIF, we failed to detect inhibition of TRIF-induced IFNβ promoter activation. TLRs are important signaling molecules particularly in cells of the immune lineage. Since immune cells are not the direct target of SARS-CoV-2 infection, it is thus not surprising that this virus lacks an ability to block TRIF signaling. Therefore there is no inconsistency with our claim that SARS-COV-2 can block early stage IFNβ and IFNλ production, with that of clinical reports that associate IFNα expression with that of severe COVID19 [72,73].

Three of the six SARS-CoV-2 genes found to repress IFNβ promoter activity (NSP1, NSP5 and NSP15) strongly suppressed both expression of the IFNβ-luciferase reporter (driven by the IFNβ promoter), but they also suppressed activation of the Renilla "control" (Fig 2A and 2B). These results support that these three genes are likely eliciting non-specific functions that are perturbing normal cellular function, and are not necessarily targeting IFN production specifically. We indeed confirmed this to be the case by performing an in-depth analysis of NSP1 function, which viral gene that acts to inhibit cellular-derived protein translation (Figs 8–10), was confirmed independently by others [62,65]. NSP5 was reported to function as a papain-like protease [36,74] and is largely responsible for cleavage of the polyprotein. NSP15 is a viral endoribonuclease and presumably is required for processing of the large viral RNA into subgenomic transcripts [64]. Each of these three viral proteins exert biological activities that are likely to disrupt regular cellular homeostasis. Their ability to block IFNβ-production is therefore not specific.

A second outcome of this study is that individual SARS-CoV-2 viral genes do not substantially suppress the activation of ISGs by added Interferons. The picture is more complex when cells are infected with SARS-CoV-2. Here, we measured the effects of individually infected cells and bystander cells by immunocytochemistry. The infected cells mostly lost pSTAT1 in the nucleus, while bystander cells maintain their interferon response (Fig 8). But as shown in S6 Fig, ISG expression and the initiation of an antiviral response requires only minute amounts of IFN-I, raising the possibility that despite that SARS-CoV-2 can diminish pSTAT1 activation, enough residual amounts must remain to allow robust IFN-I signaling to prevail. Fig 10 shows that NSP1 is responsible for not only shut-down of protein translation but also for the sharp reduction of pSTAT1. Still, at least by measuring mRNA level, ISG expression in HEK-293T cells was unaltered by NSP1, despite > 90% transfection efficiency for these cells. Interestingly, induction of ISG transcription by IFN-I was also observed upon infection with SARS-CoV-2, to the same level as non-infected cells. Furthermore, SARS-CoV-2 infected cells, in our assays remained sensitive to antiviral activity as demonstrated by the reduction in SARS-CoV-2-E gene expression (Fig 8F) and viral titer (Fig 1). Our results support a model where during early SARS-CoV-2 infection, the virus on one hand is very successful in blocking IFNβ and IFNλ production, but nevertheless infected cells remain sensitive to the effects of added interferon.

Finally, herein we have identified SARS-CoV-2 ORF7b as a potent inhibitor of MAVS-induced IFN-I and IFN-III production, mediated via either RIG-I or MDA5 (Fig 11). This was not found in some other viral screens with exception of indirect implication in a proteomic study [75], where ORF7b was found to co-precipitate with MAVS. Our finding may relate to the mild clinical pathology associated with the SARS-CoV-2 strain (Δ382) first identified in Singapore and Taiwan, which harbors a mutation in ORF7b and a functional deletion of ORF8

[68,69]. The possible contribution of deleted ORF8 in this mutant strain to reduced pathology is unlikely. Serological evidence indicates that an immune response is mounted against ORF8 (along with Spike protein) suggesting a possible evolutionary advantage to the virus after the deletion of ORF8 [76]. Furthermore, ORF8 is truncated (Q27stop) in the SARS-CoV-2 alpha variant B.1.1.7, which renders this protein inactive despite the high pathogenicity of this variant [77]. Our data presented here supports rather a rational counter-explanation, in that ORF7b contributes to the virus's ability to block IFNβ and IFNλ production, which is lost in Δ382 SARS-CoV-2.

In conclusion, our work supports a multi-gene viral-mediated blockade of MAVS-induced IFN production corresponding to that of early stages SARS-CoV-2 infection of lung cells. This blockade in IFN production provides the virus with an evolutionary advantage to remain in "stealth mode", in order to delay immune detection. Of practical application, our findings support that IFN-Is can be useful to treat early-stage SARS-CoV-2 infection. Additionally, IFN-treatment could have prophylactic application for identified individuals exposed to SARS-CoV-2 patients, where the chain of disease transmission can be curtailed or perhaps even stopped. It is not at all clear however if IFN therapy has clinical application for patients with well-established infection, where the host IFNαs are endogenously being produced. A possible exception to this counter argument however, is that for a large subset (~10%) of patients with life-threatening COVID-19, neutralizing antibodies to IFN-Is (particularly IFNα and IFNω) have been detected in patient serum, this which is almost never found in the general population, and which almost certainly is blocking the potent anti-viral effects of IFN-Is, hence contributing to the severe pathology of these patients [78]. For this subset of patients with life-threatening COVID-19, we suggest that therapeutic intervention with IFNβ may have life-saving effects, particularly because added IFNβ provides a robust anti-viral response to SARS-CoV-2 cells (Fig 1) and due to its divergent sequence to other IFN-Is, should escape IFNα/IFNω antibody-mediated neutralization.

## Supporting information

**S1 Fig. Expression levels of SARS-CoV-2 viral transcripts after infection and transfection.** **(A)** Graphical map of SARS-CoV-2 (NCBI) shows the relative positioning of SARS-CoV-2 genes. The polygenic ORF1a and ORF1b (ORF1ab) transcripts encode genes NSP1-NSP11 and NSP1-NSP16, respectively. **(B)** RNAseq data from SARS-CoV-2 infected Calu3 or VERO E6 cells (published by Finkel el. al., Nature 589:125, 2021) using normalized RPKM (Reads Per Kilobase Million) compared to qPCR data from HEK-293T cells transfected with individual SARS-CoV-2 expressing genes. The transfected viral genes included a downstream IRES and puromycin gene. Thus qPCR measurements for each viral gene was performed indirectly detecting puromycin. Small genes (ORF6 and ORF7b) were not added to the RNAseq data due to small transcript size and lack of robust read numbers. In both RNAseq and qPCR studies, gene expression was expressed as fold-change ratio in relation to the housekeeping gene, HPRT1.
(PDF)

**S2 Fig. Viral protein expression in HEK-293T cells.** Verification of viral protein expression —HEK-293T cells were transfected with vectors encoding viral proteins fused to a 2xStrep tag. 24 hours post transfection cells were collected and analyzed by western blot **(A)** or FACS **(B, C)**, using specific strep tag antibody. Samples were compared to the not-transfected control. **(C)** Quantitative analysis of the GeoMean intensity obtained in **(B)**. The dashed line indicates background signal, as determined by the non-transfected control. *: The two last panels on the right-hand-side in (A) are from the same membrane but with irrelevant intervening samples

removed.
(PDF)

**S3 Fig. TRIF induced IFNβ-luciferase promotor activation is not suppressed by SARS-CoV-2 genes.** HEK-293T cells were transfected with IFNβ promotor-Firefly luciferase, Renilla luciferase, TRIF and a SARS-CoV-2 viral gene (or control). 'No trigger' refers to cells transfected with Firefly and Renilla only; 'Trigger only' refers to cells transfected with all components but the viral gene. 24 hours post transfection luciferase activity was measured. Firefly **(A)** and Renilla **(B)** activities are presented separately. Normalized activity of Firefly/Renilla, also normalized with basal activity (no trigger), is presented in **(C)**. Data presented are means calculated based on four independent experiments and their standard error.
(PDF)

**S4 Fig. IFNα and IFNγ transcription levels are not altered by MAVS or SARS-CoV-2 genes.** HEK-293T cells were transfected with MAVS and a SARS-CoV-2 viral gene (or control). 24 hours post transfection transcript levels were analyzed by qPCR for expression of IFNα2, IFNα4, IFNα6, IFNα10 and IFNγ. The data presented are expression levels normalized to the housekeeping gene HPRT1 (ΔCT). Data presented are means of 2–4 independent experiments and their standard error.
(PDF)

**S5 Fig. Interferon stimulated gene (ISG) expression screen by qPCR.** HEK-293T cells were transfected with a viral gene. 24 hours post transfection cells were treated with low concentration (10 pM) IFNβ for additional 24 hours. Transcript levels were analyzed by qPCR, and normalized with the housekeeping gene HPRT1 (ΔCT). The GFP control was either treated (T) or not treated (NT; marked in black) with IFNβ.
(PDF)

**S6 Fig. STAT1 KnockDown (KD) does not affect IFN induced gene expression or antiviral activity. (A)** STAT1 KD and control HeLa cells were treated with 2nM IFNβ for the indicated time points and STAT protein levels were assessed by western blot. **(B)** Quantification of (A). **(C)** STAT1 KD and control HeLa cells were treated with serial dilutions of IFNβ (starting with 500pM). 4 hours later, cells were infected with Vesicular Stomatitis Virus (VSV) and after a further 18 hours, the plate was stained for cell density using crystal violet. **(D)** STAT1 KD and control HeLa cells were treated with 2nM IFNβ for 8 or 24 hours and then analyzed by qPCR. The data presented show the relative fold-change expression levels in relation to untreated cells after normalization with HPRT1. NT: Not treated.
(PDF)

**S7 Fig. Transfection efficiency of SARS-CoV-2 genes. (A)** HEK-293T cells were transfected with MAVS and a viral gene. 24 hours post transfection transcript levels were analyzed by qPCR (see Fig 4). Note: Some of data used to derive S1B Fig overlaps with data used to generate the graph shown here. **(B)** HEK- 293T cells were transfected with a viral gene. 24 hours post transfection cells were treated with 100 pM IFNβ for additional 24 hours and analyzed by qPCR (see Fig 6). All viral genes harbors a common downstream bicistronic puromycin resistance gene, which serves to evaluate transfection efficiency by qPCR. The data presented here are puromycin expression levels normalized with the housekeeping gene HPRT1 (ΔCT) of the individual repeats.
(PDF)

**S1 Table. Primers used for qPCR analyses in this paper.**
(XLSX)

## Acknowledgments

We gratefully acknowledge the help from Nevan Krogan (UCSF) for the provision of cDNA expression plasmids encoding the SARS-CoV-2 viral proteins; Ghil Jonah and Dikla Levi (Bacteriology & Genomic repository Unit, Weizmann Institute) for amplification and dissemination of the "Krogan constructs"; Noam Stern-Ginossar (Weizmann) for providing her RNASeq data of SARS-CoV-2 infected cells and also to Andy Pekosz (Johns Hopkins) for the provision of the anti-N antibody, Ashley Mansell (Hudson Institute) for the Flag-tagged MDA5 construct and Takashi Fujita (Kyoto University) for the Flag-tagged RIG-IN construct and Rongtuan Lin (McGill University) for the pGL3-IFNβ construct. We thank Paul Hertzog (Hudson Institute) for provision of IFNβ. The University of Melbourne acknowledges the support of Melbourne Health, through its Victorian Infectious Diseases Reference Laboratory at the Doherty Institute, in providing our laboratory with isolated SARS-CoV-2 material.

## Author Contributions

**Conceptualization:** Maya Shemesh, Turgut E. Aktepe, Gregory W. Moseley, Jason M. Mackenzie, Gideon Schreiber, Daniel Harari.

**Data curation:** Maya Shemesh, Turgut E. Aktepe, Joshua M. Deerain, Julie L. McAuley, Michelle D. Audsley, Cassandra T. David, Victoria Urin, Gregory W. Moseley, Jason M. Mackenzie, Gideon Schreiber, Daniel Harari.

**Formal analysis:** Maya Shemesh, Turgut E. Aktepe, Joshua M. Deerain, Julie L. McAuley, Michelle D. Audsley, Cassandra T. David, Victoria Urin, Gregory W. Moseley, Jason M. Mackenzie, Gideon Schreiber, Daniel Harari.

**Funding acquisition:** Damian F. J. Purcell, Gregory W. Moseley, Jason M. Mackenzie, Gideon Schreiber.

**Investigation:** Maya Shemesh, Turgut E. Aktepe, Joshua M. Deerain, Julie L. McAuley, Michelle D. Audsley, Cassandra T. David, Victoria Urin, Gregory W. Moseley, Jason M. Mackenzie, Gideon Schreiber, Daniel Harari.

**Methodology:** Maya Shemesh, Turgut E. Aktepe, Joshua M. Deerain, Julie L. McAuley, Michelle D. Audsley, Cassandra T. David, Victoria Urin, Rune Hartmann, Gregory W. Moseley, Jason M. Mackenzie, Gideon Schreiber, Daniel Harari.

**Project administration:** Damian F. J. Purcell, Gregory W. Moseley, Jason M. Mackenzie, Gideon Schreiber, Daniel Harari.

**Resources:** Damian F. J. Purcell, Rune Hartmann, Gregory W. Moseley, Jason M. Mackenzie, Gideon Schreiber.

**Supervision:** Damian F. J. Purcell, Gregory W. Moseley, Jason M. Mackenzie, Gideon Schreiber.

**Validation:** Maya Shemesh, Turgut E. Aktepe, Gregory W. Moseley, Jason M. Mackenzie, Gideon Schreiber, Daniel Harari.

**Visualization:** Maya Shemesh, Turgut E. Aktepe, Jason M. Mackenzie, Gideon Schreiber, Daniel Harari.

**Writing – original draft:** Maya Shemesh, Turgut E. Aktepe, Gregory W. Moseley, Jason M. Mackenzie, Gideon Schreiber, Daniel Harari.

**Writing – review & editing:** Maya Shemesh, Turgut E. Aktepe, Julie L. McAuley, Gregory W. Moseley, Jason M. Mackenzie, Gideon Schreiber, Daniel Harari.

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
