## [Decision Letter · Decision Letter 0]

30 Apr 2021

Dear Dr Harari,

Thank you very much for submitting your manuscript "SARS-CoV-2 suppresses IFNβ production mediated by NSP1, 5, 6, 15, ORF6 and ORF7b but does not suppress the effects of added interferon." for consideration at PLOS Pathogens. As with all papers reviewed by the journal, your manuscript was reviewed by members of the editorial board and by several independent reviewers. In light of the reviews (below this email), we would like to invite the resubmission of a significantly-revised version that takes into account the reviewers' comments.

Both reviewers found the study interesting, important and thorough. Both had some good suggestions that will strengthen the conclusions that can be drawn from the work.

We cannot make any decision about publication until we have seen the revised manuscript and your response to the reviewers' comments. Your revised manuscript is also likely to be sent to reviewers for further evaluation.

Sincerely,

Andrew Pekosz, Ph.D.

Section Editor

PLOS Pathogens

Andrew Pekosz

Section Editor

PLOS Pathogens

Kasturi Haldar

Editor-in-Chief

PLOS Pathogens

orcid.org/0000-0001-5065-158X

Michael Malim

Editor-in-Chief

PLOS Pathogens

orcid.org/0000-0002-7699-2064

Both reviewers found the study interesting, important and thorough. Both had some good suggestions that will strengthen the conclusions that can be drawn from the work.

Reviewer's Responses to Questions

**Part I - Summary**

Reviewer #1: (No Response)

Reviewer #2: ‘SARS-CoV-2 suppresses IFNβ production 1 mediated by NSP1, 5, 6, 15, ORF6 and ORF7b but does not suppress the effects of added interferon’ by Maya Shemesh et al provides a detailed examination of 27 SARS-CoV-2 genes on interferon production. The authors examine MAVS and TRIF mediated IFNb signaling pathways with all genes and also examine a subset of genes and their ability to inhibit ISG expression. Six SARS-CoV-2 genes were found to block IFNb signaling although nsp1 was shown to have a global effect on expression. The manuscript is very thorough and the authors have taken care to consider previous work in this area and put their results into the context of previous studies. The summary in table 1 is very helpful. Overall, this is an extensive and detailed analysis but the heavy reliance on transfection and RNA data and number of previous publications in this area are serious weaknesses.

**Part II – Major Issues: Key Experiments Required for Acceptance**

Reviewer #1: (No Response)

Reviewer #2: Expression data for transfected genes as presented is odd/incomplete. It is far preferable to show protein by western blot.

Figure 5 – It is important to consider that the cumulative action of several SAR2 proteins could suppress ISG expression when a single gene product cannot.

Figure 5 – needs a positive control showing that they can block ISG induction from this level of IFNb treatment, it is possible that the system is being overwhelmed.

**Part III – Minor Issues: Editorial and Data Presentation Modifications**

Reviewer #1: I enjoyed reading this paper. The data are valuable for understanding not only SARS-CoV-2 biology but also potential therapeutic approaches. The experiments are rigorous and the conclusions are justified by the data presented. I have some minor comments:

The authors indirectly show a decrease in virus-infected cells after addition of IFNa by qPCR. It would be worth repeating this with more direct evidence of decrease in virus.

Since the authors show that ifnb is upregulated by MAVS, this assay should be re-done with both IFNA and IFNB.

Reviewer #2: Figure 1 – The black and green bars coloration is explained in the figure legend, should also indicate what blue vs gray means in the legend.

Figure 3 legend – should refer to parts A, B and C

Line 339 – “do not directly assess the amount of IFNb protein…” – these assays don’t measure protein at all. Please use more precise phrasing.

Why not do an ELISA to measure IFNb protein levels?

Line 360-361 – Should say that 4 ISGs were measured?

Why the switch from IFNb to IFNa stimulation between figures 5 and 6?

Figure 7 labels – what is RbStat1?

Figure 9 legend – please mention the star present on the nsp1 images

Line 571 – “SARS-CoV-2 does not suppress IFN-activation” – should this read ISG activation?

PLOS authors have the option to publish the peer review history of their article (what does this mean?). If published, this will include your full peer review and any attached files.

Reviewer #1: No

Reviewer #2: No
---

## [Editor Report · Decision Letter 1]

14 Jul 2021

Dear Dr Harari,

We are pleased to inform you that your manuscript 'SARS-CoV-2 suppresses IFNβ production mediated by NSP1, 5, 6, 15, ORF6 and ORF7b but does not suppress the effects of added interferon.' has been provisionally accepted for publication in PLOS Pathogens.

Best regards,

Andrew Pekosz, Ph.D.

Section Editor

PLOS Pathogens

Andrew Pekosz

Section Editor

PLOS Pathogens

Kasturi Haldar

Editor-in-Chief

PLOS Pathogens

orcid.org/0000-0001-5065-158X

Michael Malim

Editor-in-Chief

PLOS Pathogens

orcid.org/0000-0002-7699-2064
---

## [Editor Report · Acceptance letter]

5 Aug 2021

Dear Dr Harari,

We are delighted to inform you that your manuscript, "SARS-CoV-2 suppresses IFNβ production mediated by NSP1, 5, 6, 15, ORF6 and ORF7b but does not suppress the effects of added interferon.," has been formally accepted for publication in PLOS Pathogens.

Best regards,

Kasturi Haldar

Editor-in-Chief

PLOS Pathogens

orcid.org/0000-0001-5065-158X

Michael Malim

Editor-in-Chief

PLOS Pathogens

orcid.org/0000-0002-7699-2064